# Eliminating Catastrophic Overfitting Via Abnormal Adversarial Examples Regularization

**Runqi Lin**   **Chaojian Yu**   **Tongliang Liu**[*]

Sydney AI Centre, The University of Sydney

{rlin0511, chyu8051, tongliang.liu}@sydney.edu.au

## Abstract

Single-step adversarial training (SSAT) has demonstrated the potential to achieve both efficiency and robustness. However, SSAT suffers from catastrophic overfitting (CO), a phenomenon that leads to a severely distorted classifier, making it vulnerable to multi-step adversarial attacks. In this work, we observe that some adversarial examples generated on the SSAT-trained network exhibit anomalous behaviour, that is, although these training samples are generated by the inner maximization process, their associated loss decreases instead, which we named abnormal adversarial examples (AAEs). Upon further analysis, we discover a close relationship between AAEs and classifier distortion, as both the number and outputs of AAEs undergo a significant variation with the onset of CO. Given this observation, we re-examine the SSAT process and uncover that before the occurrence of CO, the classifier already displayed a slight distortion, indicated by the presence of few AAEs. Furthermore, the classifier directly optimizing these AAEs will accelerate its distortion, and correspondingly, the variation of AAEs will sharply increase as a result. In such a vicious circle, the classifier rapidly becomes highly distorted and manifests as CO within a few iterations. These observations motivate us to eliminate CO by hindering the generation of AAEs. Specifically, we design a novel method, termed *Abnormal Adversarial Examples Regularization* (AAER), which explicitly regularizes the variation of AAEs to hinder the classifier from becoming distorted. Extensive experiments demonstrate that our method can effectively eliminate CO and further boost adversarial robustness with negligible additional computational overhead. Our implementation can be found at https://github.com/tmllab/2023_NeurIPS_AAER.

## 1   Introduction

In recent years, deep neural networks (DNNs) have demonstrated impressive performance in various decision-critical domains, such as autonomous driving [25, 12], face recognition [31, 3] and medical imaging diagnosis [8]. However, DNNs were found to be vulnerable to adversarial examples [35, 9, 16]. Although these adversarial perturbations are imperceptible to human eyes, they can lead to a completely different prediction in DNNs. To this end, many adversarial defence strategies have been proposed, such as pre-processing techniques [13], detection algorithms [27], verification and provable defence [20, 44], and adversarial training (AT) [11, 26, 46]. Among them, AT is considered to be the most effective method against adversarial attacks [2, 4].

Despite the notable progress in improving model robustness, the standard multi-step AT significantly increases the computational overhead due to the iterative steps of forward and backward propagation [26, 40, 41, 42, 48, 47]. In light of this, several works have attempted to use single-step adversarial training (SSAT) as a more efficient alternative to achieve robustness. Unfortunately, a

---

[*]Corresponding author

37th Conference on Neural Information Processing Systems (NeurIPS 2023).

serious problem catastrophic overfitting (CO) has been identified in SSAT [39], manifesting as a sharp decline in the model's robust accuracy against multi-step adversarial attacks, plummeting from a peak to nearly 0% within a few iterations, as shown in Figure 1. This intriguing phenomenon has been widely investigated and prompted numerous efforts to resolve it. Recently, Kim [21] pointed out that the SSAT-trained classifiers are typically accompanied by highly distorted decision boundaries, which will lead to the model manifestation as CO. However, the underlying process of the classifier's gradual distortion, as well as the factor inducing rapid distortion, remains unclear.

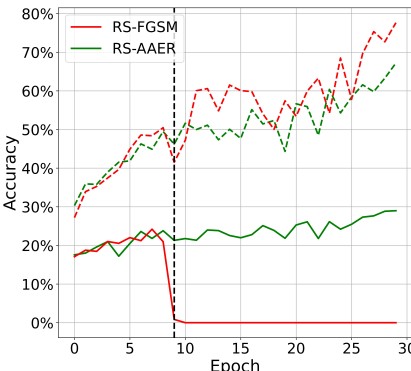

Figure 1. The test accuracy of RS-FGSM [39] (red line) and RS-AAER (green line) with 16/255 noise magnitude. The dashed and solid lines denote natural and robust (PGD-7-1) accuracy, respectively. The dashed black line corresponds to the 9th epoch, which is the point that RS-FGSM occurs CO.

In this study, we identify some adversarial examples generated by the distorted classifier exhibiting anomalous behaviour, wherein the loss associated with them decreases despite being generated by the inner maximization process. We refer to these anomalous training samples as abnormal adversarial examples (AAEs). Upon further investigation of the training process, we observe that both the number and outputs of AAEs undergo a significant variation during CO. This observation suggests a strong correlation between the variation of AAEs and the gradually distorted classifier. By utilizing AAEs as the indicator, we re-evaluate the process of SSAT and uncover that the classifier already exhibits slight distortions even before the onset of CO, which is evidenced by the presence of few AAEs. To make matters worse, directly optimizing the model based on these AAEs will further accelerate the distortion of the decision boundaries. Furthermore, in response to this more distorted classifier, the variation in AAE will dramatically increase as a result. This interaction leads to a vicious circle between the variation of AAEs and the decision boundaries distortion, ultimately leading to the model rapidly manifesting as CO. All these atypical findings raise a question:

*Can CO be prevented by hindering the generation of abnormal adversarial examples?*

To answer the above question, we design a novel method, called *Abnormal Adversarial Examples Regularization* (AAER), which prevents CO by incorporating a regularizer term designed to suppress the generation of AAEs. Specifically, to achieve this objective, AAER consists of two components: the number and the outputs variation of AAEs. The first component identifies and counts the number of AAEs in the training samples through anomalous loss decrease behaviour. The second component calculates the outputs variation of AAEs by combining the prediction confidence and logits distribution. Subsequently, AAER explicitly regularizes both the number and the outputs variation of AAEs to prevent the model from being distorted. It is worth noting that our method does not involve any extra example generation or backward propagation processes, making it highly efficient in terms of computational overhead. Our major contributions are summarized as follows:

- We identify a particular behaviour in SSAT, in which some AAEs generated by the distorted classifier have an opposite objective to the maximization process, and their number and outputs variation are highly correlated with the classifier distortion.

- We discover that the classifier exhibits initial distortion before CO, manifesting as a small number of AAEs. Besides, the model decision boundaries will be further exacerbated by directly optimizing the classifier on these AAEs, leading to a further increase in their number, which ultimately manifests as CO within a few iterations.

- Based on the observed effect, we propose a novel method - *Abnormal Adversarial Examples Regularization* (AAER), which explicitly regularizes the number and outputs variation of AAEs to hinder the classifier from becoming distorted. We evaluate the effectiveness of our method across different adversarial budgets, adversarial attacks, datasets and network architectures, showing that our proposed method can consistently prevent CO even with extreme adversaries and boost robustness with negligible additional computational overhead.

## 2 Related Work

### 2.1 Adversarial Training

DNNs are known to be vulnerable to adversarial attacks [35], and AT has been demonstrated to be the most effective defence method [2]. AT is generally formulated as a min-max optimization problem [26, 4]. The inner maximization problem tries to generate the strongest adversarial examples to maximize the loss, and the outer minimization problem tries to optimize the network to minimize the loss on adversarial examples, which can be formalized as follows:

$$\min_{\theta} \mathbb{E}_{(x,y)\sim\mathcal{D}} \left[ \max_{\delta\in\Delta} \ell(x + \delta, y; \theta) \right],\qquad(1)$$

where $(x, y)$ is the training dataset from the distribution $D$, $\ell(x, y; \theta)$ is the loss function parameterized by $\theta$, $\delta$ is the perturbation confined within the boundary $\epsilon$ shown as: $\Delta = \{\delta : \|\delta\|_p \leq \epsilon\}$.

### 2.2 Catastrophic Overfitting

Since the intriguing phenomenon of CO was identified [39], there has been a line of work trying to explore and mitigate this problem. [39] first suggested using a random initialization and early stopping to avoid CO. Furthermore, [36] empirically showed that using a dynamic dropout schedule can avoid early overfitting to adversarial examples, and [6, 45] found that incorporating a stronger data augmentation is effective in avoiding CO. Another alternative approach imports partial multi-step AT, for example, [38] periodically trained the model on natural, single-step and multi-step adversarial examples, and [37] built a regularization term by comparing with the multi-step adversarial examples.

However, the above methods have not provided a deeper insight into the essence of CO. Separate works found that CO is closely related to anomalous gradient updates. [24] constrained the training samples to a carefully extracted subspace to avoid abrupt gradient growth. [10] ignored the small gradient adversarial perturbations to mitigate substantial weight updates in the network. [15] proposed an instance-adaptive SSAT approach where the perturbation size is inversely proportional to the gradient. [29] leveraged the latent representation of gradients as the adversarial perturbation to compensate for local linearity. [34] introduced a relaxation term to find more suitable gradient directions by smoothing the loss surface. [1] proposed a regularization term to avoid the non-linear surfaces around the samples. More recently, [21] introduced a new perspective that CO is a manifestation of highly distorted decision boundaries. Accordingly, they proposed to reduce the perturbation size for the already misclassified adversarial examples.

Unfortunately, the aforementioned methods tend to either suffer from CO with strong adversaries or significantly increase the computational overhead. In this work, we delve into the interaction between AAEs and distorted decision boundaries, revealing a close relationship between them. Based on this insight, we propose a novel approach, AAER, that eliminates CO by explicitly hindering the generation of AAEs, thereby achieving both efficiency and robustness.

## 3 Methodology

In this section, we first define the abnormal adversarial example (AAE) and show how their numbers change throughout the training process (Section 3.1). We further compare the outputs variation of normal adversarial examples (NAEs) and AAEs and find that their outputs exhibit significantly different behaviour after CO (Section 3.2). Building upon our observations, we propose a novel regularization term, *Abnormal Adversarial Examples Regularization* (AAER), that uses the number and outputs variation of AAEs to explicitly suppress the generation of them to eliminate catastrophic overfitting (CO) (Section 3.3).

### 3.1 Definition and Counting of AAE

Adversarial training employs the most adversarial data to reduce the sensitivity of the network's outputs w.r.t. adversarial perturbation of the natural data. Consequently, the inner maximization process is expected to generate effective adversarial examples that maximize the classification loss. As empirically demonstrated by [21], the decision boundaries of the classifier become highly distorted

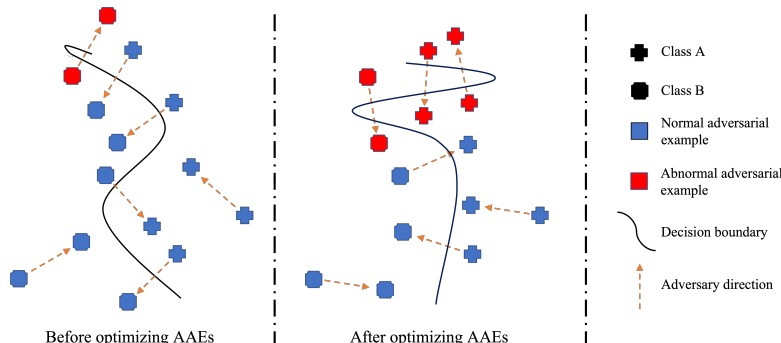

Figure 2. A conceptual diagram of the classifier's decision boundary and training samples. The training samples belonging to NAE (blue) can effectively mislead the classifier, while AAE (red) cannot. The left panel shows the decision boundary before optimizing AAEs, which only has a slight distortion. The middle panel shows the decision boundary after optimizing AAEs, which exacerbates the distortion and generates more AAEs.

after the occurrence of CO. In this study, we find that after adding the adversarial perturbation generated by the distorted classifier, the loss of certain training samples unexpectedly decreases. This particular behaviour is illustrated in Figure 2, we can observe that the NAEs (blue) can either lead to the model misclassifications or position themselves closer to the decision boundary after the inner maximization process. In contrast, the AAEs (red) will be located further away from the decision boundary and fail to mislead the classifier after adding the perturbation generated by the distorted classifier. Therefore, we introduce the following formula to define AAEs:

$$
\begin{aligned}
\delta &= \mathrm{sign}\left(\nabla_{x+\eta}\ell(x+\eta, y; \theta)\right), \\
x^{AAE} &\stackrel{def}{=} \ell\left(x+\eta, y; \theta\right) > \ell\left(x+\eta+\delta, y; \theta\right),
\end{aligned}
\tag{2}
$$

where $\eta$ is the random initialization.

Next, we observe the variation in the number of AAEs throughout the model training process, and the corresponding statistical results are presented in Figure 3 (left). It can be observed that before the occurrence of CO, a small number of AAEs already existed, indicating the presence of slight initial distortion in the classifier. To further validate this point, we visualize the loss surface of both AAEs and NAEs using the method proposed by [23] as shown in Figure 4 (left and middle). It's evident that before CO, the classifier showcases a more nonlinear loss surface around AAEs in comparison to NAEs. This empirical observation strongly suggests that the generation of AAEs is directly influenced by the distorted classifier.

Besides, the number of AAEs experiences a dramatic surge during CO occurrences. For example, the number of AAEs (red line) exploded 19 times at the onset of CO (9th epoch), as shown in Figure 3 (left). Importantly, this rapid increase in the AAEs number implies a continuous deterioration in the classifier's boundaries, which in turn leads to a further increase in their number. The number of AAEs reaches its peak at the 10th epoch, surging to approximately 66 times than that before CO. To meticulously analyze the interaction between AAEs and CO, we delve into their relationship at the iteration level, as shown in Figure 4 (right). We can observe that the robustness accuracy from peak sharply drops to nearly 0% within 18 iterations, simultaneously, the number of AAEs rises from 0 to 70. Remarkably, the trends in robustness accuracy and the number of AAEs display a completely opposite relationship, suggesting a vicious cycle between optimizing AAEs and CO. Lastly, the number of AAEs consistently maintains a high level until the end of the training. Given this empirical and statistical observation, we can infer that there is a close correlation between the number of AAEs and the CO phenomenon, which also prompts us to wonder (Q1): *whether CO can be mitigated by reducing the number of abnormal adversarial examples.*

### 3.2 Outputs Variation of NAE and AAE

The above observations indicate the close relationship between CO and AAEs. In this part, we further analyze the outputs variation of AAEs during CO. Specifically, we discover that CO has a

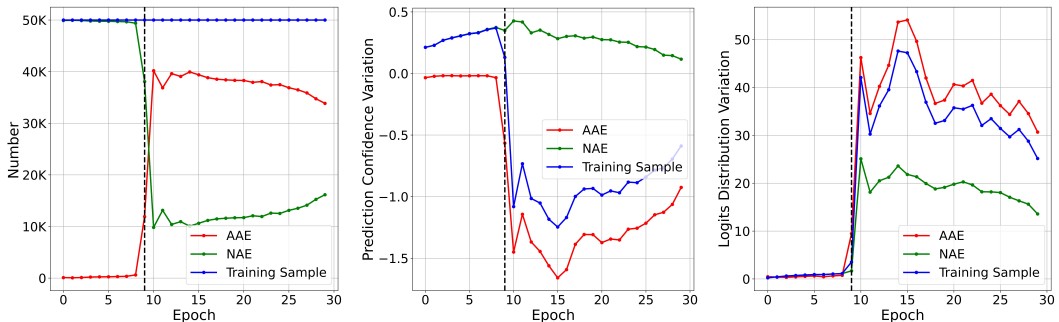

Figure 3. The number, the variation of prediction confidence and logits distribution (from left to right) for NAEs, AAEs and training samples in RS-FGSM with 16/255 noise magnitude. The dashed black line corresponds to the 9th epoch, which is the point that the model occurs CO.

significant impact on both the prediction confidence and the logits distribution of AAEs. To quantify the variation in prediction confidence, we utilize the cross-entropy (CE) to calculate the change in loss during the inner maximization process, which is formulated as follows:

$$\ell\left(x + \eta + \delta, y; \theta\right) - \ell\left(x + \eta, y; \theta\right). \tag{3}$$

We investigate the prediction confidence of NAEs and AAEs variation during the model training. From Figure 3 (middle), we can observe that the change in prediction confidence of NAEs is consistently greater than 0, indicating that their lead to a worse prediction in the classifier. On the contrary, this variation in AAEs is atypical negative implying that the associated adversarial perturbation has an unexpected opposite effect. Moreover, we delve into the impact of CO on the variation in prediction confidence. Before CO, we note a slight negative variation in the AAEs' prediction confidence, which has an insignificant impact on all training samples (blue line). However, during CO, the prediction confidence of AAEs undergoes a rapid and substantial drop, reaching a decline of 17 times at the 9th epoch. After CO, the prediction confidence of AAEs is 43 times (10th epoch) smaller than before and significantly impacts all training samples.

In addition to the inability to mislead the classifier, the logits distribution of AAE is also disturbed during the CO process. To analyze the variation in logits distribution, we employ the Euclidean (L2) distance to quantify the impact of adversarial perturbation, which is formulated as follows:

$$\|f_\theta\left(x + \eta + \delta\right) - f_\theta\left(x + \eta\right)\|_2^2, \tag{4}$$

where $f_\theta$ is the DNN classifier parameterized by $\theta$ and $\|\cdot\|_2^2$ is the L2 distance.

The logits distribution variation of both NAEs and AAEs are illustrated in Figure 3 (right). Comparing the logits distribution variation between NAEs and AAEs, we can find that their magnitudes are similar before CO. However, it becomes evident that the logits distribution variation of AAEs increases dramatically during CO, being 13 times larger than before. After further optimization on AAEs, the variation in logits distribution reaches the peak, approximately 62 times larger than before. This observation highlights that even a small adversarial perturbation can cause a substantial variation in the logits distribution, this phenomenon typically happens on the highly distorted decision boundaries. Additionally, it's worth noting that the increase in logits distribution variation for NAEs (green line) occurs one epoch later than that of AAEs, indicating that the primary cause of decision boundary distortion lies within the AAEs. In other words, directly optimizing the network using these AAEs exacerbates the distortion of decision boundaries, resulting in a significant change in the logits distribution for NAEs. Even after CO, the logits distribution variance of AAEs remains twice as large as NAEs. The significant difference between NAEs and AAEs in the variation of prediction confidence and logits distribution inspires us to wonder (Q2): *whether CO can be mitigated by constraining the outputs variation of abnormal adversarial examples.*

### 3.3  Abnormal Adversarial Examples Regularization

Recognizing the strong correlation between CO and AAEs, we first attempt a passive approach by removing AAEs and training solely on NAEs. This simple approach demonstrates the capability

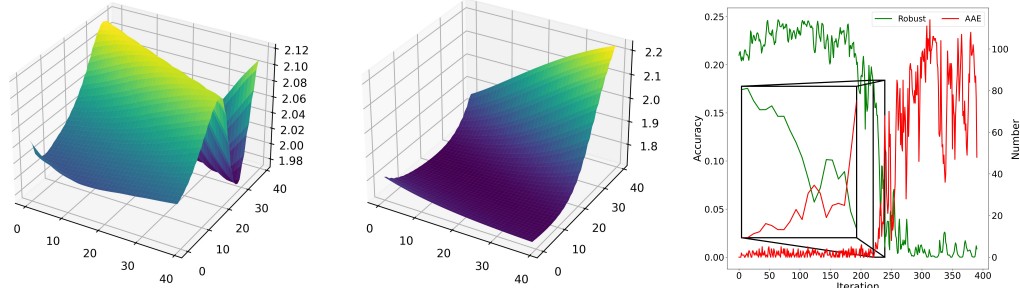

Figure 4. Left/Middle panel: The visualization of AAEs/NAEs loss surface before CO (8th epoch). Right panel: The number of AAEs and the test robustness within each iteration at CO (9th epoch). The green and red lines represent the robust accuracy and number of AAEs, respectively.

to delay the onset of CO, thereby confirming that direct optimization of AAEs will accelerate the classifier's distortion. However, it is important to note that the generation of AAEs is caused by the distorted classifier. Passively removing AAEs cannot provide the necessary constraints to promote smoother classifiers, thereby only delaying the onset of CO but not preventing it.

To truly relieve this problem, we design a novel regularization term, *Abnormal Adversarial Examples Regularization* (AAER), which aims to hinder the classifier from becoming distorted by explicitly reducing the number and constraining the outputs variation of AAEs. Specifically, part (*i*) categorizes the training samples into NAEs and AAEs according to the definition in Eq. 2, and then penalizes the number of AAEs. The AAEs' outputs variation is simultaneously constrained by part (*ii*) prediction confidence and part (*iii*) logits distribution. In terms of prediction confidence, we penalize the anomalous variation in AAEs that should not be negative during the inner maximization process, which is formalized as follows:

$$AAE\_CE = \frac{1}{n} \sum_{i=1}^{n} \left( \ell \left( x_i^{AAE} + \eta, y_i; \theta \right) - \ell \left( x_i^{AAE} + \eta + \delta, y_i; \theta \right) \right), \tag{5}$$

where $n$ is the number of abnormal adversarial examples.

For logits distribution, we first calculate the logits distribution variation of AAEs and NAEs separately, as shown in Eq. (6) and Eq. (7):

$$AAE\_L2 = \frac{1}{n} \sum_{i=1}^{n} \left( \| f_\theta \left( x_i^{AAE} + \eta + \delta \right) - f_\theta \left( x_i^{AAE} + \eta \right) \|_2^2 \right); \tag{6}$$

$$NAE\_L2 = \frac{1}{m-n} \sum_{j=1}^{m-n} \left( \| f_\theta \left( x_j^{NAE} + \eta + \delta \right) - f_\theta \left( x_j^{NAE} + \eta \right) \|_2^2 \right), \tag{7}$$

where $m$ is the number of training samples.

Then, we use the logits distribution variation of NAEs as a reference to constrain the variation in AAEs. It's essential to emphasize that our optimization objective is to make the logits distribution variation of AAEs closer to that of NAEs, rather than less. To achieve this, we use the max function to limit the minimum value, which is formalized as follows:

$$Constrained\_Variation = max \left( AAE\_L2 \ (6) - NAE\_L2 \ (7), 0 \right), \tag{8}$$

where $max(,)$ is the max function.

Although Figure 3 (right) illustrates that the logits distribution variation of NAEs will significantly increase and instability after CO. However, that is a natural consequence of the highly distorted classifier which disrupted the logits distribution of NAEs. In contrast, after using the AAER to hinder the classifier from becoming distorted, the NAEs can be used as a stable standard throughout the training, as shown in Figure 5 (a:right).

Based on the optimization objectives described above, we can build a novel regularization term - AAER, which aims to suppress the AAEs by the number, the variation of prediction confidence

---
**Algorithm 1** *Abnormal Adversarial Examples Regularization* (AAER)
---
**Input:** Network $f_\theta$, epochs T, mini-batch M, perturbation radius $\epsilon$, step size $\alpha$, initialization term $\eta$.
**Output:** Adversarially robust model $f_\theta$
 1: **for** $t = 1 \ldots T$ **do**
 2:     **for** $k = 1 \ldots M$ **do**
 3:         $\delta = \alpha \cdot \text{sign}\left(\nabla_{x+\eta}\ell(x_k + \eta, y_k; \theta)\right)$
 4:         $CE = \frac{1}{m}\sum_{k=1}^{m}\ell\left(x_k + \eta + \delta, y_k; \theta\right)$
 5:         $AAER = $ Eq. (9)
 6:         $\theta = \theta - \nabla_\theta\left(CE + AAER\right)$
 7:     **end for**
 8: **end for**
---

and logits distribution, ultimately achieving the purpose of preventing CO, which is shown in the following formula:

$$AAER = \left(\lambda_1 \cdot \frac{n}{m}\right) \cdot \left(\lambda_2 \cdot AAE\_CE\ (5) + \lambda_3 \cdot Constrained\_Variation\ (8)\right), \qquad (9)$$

where $\lambda_1$, $\lambda_2$ and $\lambda_3$ are the hyperparameters to control the strength of the regularization term.

AAER can effectively hinder the generation of AAEs that are highly correlated with the distorted classifier and CO, thereby encouraging training for a smoother classifier that can effectively defend against adversarial attacks. By considering both the number and output variation of AAEs, we establish a more adaptable and comprehensive measure of classifier distortion. Importantly, our method does not require any additional generation or backward propagation processes, making it highly convenient in terms of computational overhead. The proposed algorithm AAER realization is summarized in Algorithm 1.

## 4 Experiment

In this section, we provide a comprehensive evaluation to verify the effectiveness of AAER, including experiment settings (Section 4.1), performance evaluation (Section 4.2), ablation studies (Section 4.3) and time complexity study (Section 4.4).

### 4.1 Experiment Settings

**Baselines.** We compare our method with other SSAT methods, including RS-FGSM [39], FreeAT [30], N-FGSM [6], Grad Align [1], ZeroGrad and MultiGrad [10]. We also compare our method with multi-step AT, PGD-2 and PGD-10 (PGD-20 with 32/255 noise magnitude) [26], providing a reference for the ideal performance. The results of other competing baselines, including GAT [33], NuAT [34], PGI-FGSM [17], SDI-FGSM [18] and Kim [21], can be found in Appendix F. We report both the natural and robust accuracy results of the final model, which are obtained without early stopping and using the hyperparameters provided in the official repository. Please note that for FreeAT, we did not use the subset of training samples to keep the same training epochs across different methods.

**Attack Methods.** To report the robust accuracy of models, we attack these methods using the standard PGD adversarial attack with $\alpha = \epsilon/4$ step size, 50 attack steps and 10 restarts. We also evaluate our methods on Auto Attack [5] as shown in Appendix C.

**Datasets and Model Architectures.** We evaluate our method on several benchmark datasets, including Cifar-10/100 [22], SVHN [28], Tiny-ImageNet [28] and Imagenet-100 [7]. The standard data augmentation random cropping and horizontal flipping are applied for these datasets. The settings and results on SVHN, Tiny-ImageNet and Imagenet-100 are provided in Appendix E. We use the PreactResNet-18 [14] and WideResNet-34 [43] architectures on these datasets to evaluate results. The results of WideResNet-34 can be found in Appendix D.

**Setup for Our Proposed Method.** In this work, we use the SGD optimizer with a momentum of 0.9, weight decay of $5 \times 10^{-4}$ and $L_\infty$ as the threat model. For the learning rate schedule, we use the cyclical learning rate schedule [32] with 30 epochs, which reaches its maximum learning rate

Table 1. The hyperparameter settings for Cifar-10/100 are divided by a slash. The top number represents $\lambda_2$ while the bottom number represents $\lambda_3$. Throughout all settings, $\lambda_1$ is fixed at 1.0.

| CIFAR10/100 | RS-AAER | | | | N-AAER | | | |
|---|---|---|---|---|---|---|---|---|
| | 8/255 | 12/255 | 16/255 | 32/355 | 8/255 | 12/255 | 16/255 | 32/355 |
| $\lambda_2$ | 2.5 / 3.5 | 5.0 / 3.5 | 7.0 / 6.0 | 5.75 / 5.0 | 1.5 / 1.5 | 5.0 / 3.5 | 8.5 / 6.0 | 2.75 / 3.5 |
| $\lambda_3$ | 1.5 / 1.5 | 2.75 / 2.75 | 3.25 / 2.25 | 1.5 / 0.75 | 0.15 / 0.15 | 0.55 / 0.3 | 1.5 / 0.5 | 0.75 / 0.5 |

Table 2. CIFAR10/100: Accuracy of different methods and different noise magnitudes using PreActResNet-18 under $L_\infty$ threat model. The top number is the natural accuracy (%), while the bottom number is the PGD-50-10 accuracy (%). The results are averaged over 3 random seeds and reported with the standard deviation.

| dataset | CIFAR10 | | | | CIFAR100 | | | |
|---|---|---|---|---|---|---|---|---|
| noise magnitude | 8/255 | 12/255 | 16/255 | 32/255 | 8/255 | 12/255 | 16/255 | 32/255 |
| FreeAT | 76.20 ± 1.09 | 68.07 ± 0.38 | 45.84 ± 19.07 | 61.11 ± 8.41 | 47.41 ± 0.30 | 39.84 ± 0.40 | 3.32 ± 2.48 | 26.2 ± 15.54 |
| | 43.74 ± 0.41 | 33.14 ± 0.62 | 0.00 ± 0.00 | 0.00 ± 0.00 | 22.27 ± 0.33 | 16.57 ± 0.20 | 0.00 ± 0.00 | 0.00 ± 0.00 |
| ZeroGrad | 81.60 ± 0.16 | 77.52 ± 0.21 | 79.65 ± 0.17 | 65.48 ± 6.26 | 53.83 ± 0.22 | 49.07 ± 0.14 | 50.76 ± 0.02 | 49.38 ± 1.39 |
| | 47.56 ± 0.16 | 27.34 ± 0.09 | 6.37 ± 0.23 | 0.00 ± 0.00 | 25.02 ± 0.24 | 14.76 ± 0.26 | 5.23 ± 0.09 | 0.00 ± 0.00 |
| MultiGrad | 81.65 ± 0.16 | 81.09 ± 4.67 | 82.98 ± 3.30 | 70.84 ± 4.53 | 53.11 ± 0.34 | 46.81 ± 0.51 | 46.05 ± 8.68 | 28.33 ± 6.48 |
| | 47.93 ± 0.18 | 9.95 ± 16.97 | 0.00 ± 0.00 | 0.00 ± 0.00 | 25.68 ± 0.21 | 16.56 ± 0.56 | 0.00 ± 0.00 | 0.00 ± 0.00 |
| Grad Align | 82.10 ± 0.78 | 74.17 ± 0.55 | 60.37 ± 0.95 | 25.23 ± 3.41 | 54.00 ± 0.44 | 45.83 ± 0.72 | 36.80 ± 0.10 | 15.05 ± 0.07 |
| | 47.77 ± 0.58 | 34.87 ± 1.00 | 27.90 ± 1.01 | 11.53 ± 3.23 | 25.27 ± 0.68 | 18.13 ± 0.71 | 13.77 ± 0.76 | 2.85 ± 1.34 |
| RS-FGSM | 83.91 ± 0.21 | 66.46 ± 22.80 | 66.54 ± 12.25 | 36.43 ± 7.86 | 60.29 ± 1.51 | 18.19 ± 8.51 | 11.03 ± 5.24 | 11.40 ± 8.60 |
| | 46.01 ± 0.18 | 0.00 ± 0.00 | 0.00 ± 0.00 | 0.00 ± 0.00 | 10.58 ± 13.10 | 0.00 ± 0.00 | 0.00 ± 0.00 | 0.00 ± 0.00 |
| N-FGSM | 80.48 ± 0.21 | 71.30 ± 0.12 | 62.96 ± 0.74 | 29.79 ± 3.87 | 54.92 ± 0.28 | 46.16 ± 0.13 | 37.93 ± 0.22 | 18.18 ± 4.55 |
| | 47.91 ± 0.29 | 36.23 ± 0.10 | 27.14 ± 1.44 | 8.30 ± 7.85 | 26.29 ± 0.41 | 18.75 ± 0.19 | 14.05 ± 0.07 | 0.00 ± 0.00 |
| RS-AAER | 83.83 ± 0.27 | 74.40 ± 0.79 | 64.56 ± 1.45 | 31.58 ± 1.13 | 57.71 ± 0.29 | 44.06 ± 0.93 | 33.10 ± 0.05 | 18.50 ± 1.68 |
| | 46.14 ± 0.02 | 32.17 ± 0.16 | 23.87 ± 0.36 | 10.62 ± 0.51 | 25.31 ± 0.01 | 16.41 ± 0.13 | 11.80 ± 0.17 | 4.90 ± 0.50 |
| N-AAER | 80.56 ± 0.35 | 71.15 ± 0.18 | 61.84 ± 0.43 | 27.08 ± 0.02 | 54.47 ± 0.45 | 45.98 ± 0.13 | 36.80 ± 0.14 | 16.95 ± 0.44 |
| | **48.31 ± 0.23** | **36.52 ± 0.10** | **28.20 ± 0.71** | **12.97 ± 0.57** | **26.81 ± 0.13** | **19.03 ± 0.04** | **14.31 ± 0.05** | **5.45 ± 0.14** |
| PGD-2 | 85.07 ± 0.12 | 78.97 ± 0.23 | 72.31 ± 0.40 | 48.45 ± 0.71 | 60.09 ± 0.20 | 53.46 ± 0.27 | 47.50 ± 0.28 | 31.89 ± 0.69 |
| | 45.27 ± 0.07 | 32.99 ± 0.46 | 24.32 ± 0.64 | 11.24 ± 0.40 | 24.58 ± 0.12 | 17.16 ± 0.21 | 12.69 ± 0.06 | 4.51 ± 0.21 |
| PGD-10 (20) | 80.55 ± 0.37 | 72.37 ± 0.31 | 67.20 ± 0.69 | 34.70 ± 0.67 | 55.05 ± 0.25 | 47.42 ± 0.29 | 42.39 ± 0.17 | 21.68 ± 0.18 |
| | **50.67 ± 0.40** | **38.60 ± 0.39** | **29.34 ± 0.18** | **16.10 ± 0.20** | **27.87 ± 0.12** | **20.29 ± 0.18** | **15.01 ± 0.21** | **7.39 ± 0.38** |

(0.2) when half of the epochs (15) are passed. The results obtained by the long training schedule can be found in Appendix G. We superimpose our method on two baseline methods: RS-FGSM and N-FGSM, both of which use the vanilla min-max process. For RS-AAER, we follow the settings of [39] that set step size $\alpha = 1.25 \cdot \epsilon$ and random initialization $\eta = \text{Uniform}(-\epsilon, \epsilon)$. In accordance with the N-FGSM setting suggested by [6], we set $\alpha = 1.0 \cdot \epsilon$ and $\eta = \text{Uniform}(-2 \cdot \epsilon, 2 \cdot \epsilon)$ for N-AAER. The hyperparameter settings for Cifar-10/100 are summarized in the Table 1.

## 4.2 Performance Evaluation

**CIFAR10 Results.** In Table 2 (left), we present a comparison of our proposed methods with the competing baselines. First, we can observe that CO occurs in all baseline methods except for Grad Align. However, Grad Align requires double backward propagation which notably reduces its efficiency. Compared to them, RS-AAER and N-AAER can effectively eliminate CO with all noise magnitudes and only incur negligible computational overhead. It is worth noting that our primary objective is to eliminate CO in SSAT, thus AAER will show significant performance enhancements under CO scenarios, exemplified by RS-AAER under 12, 16 and 32 noise magnitude, and N-AAER under 32 noise magnitude. Even without the CO scenario, our approach can consistently outperform the baselines under all experimental conditions. Additionally, we implemented Vanilla-AAER in Appendix A to further demonstrate the effectiveness of our method. For a fair comparison, we also

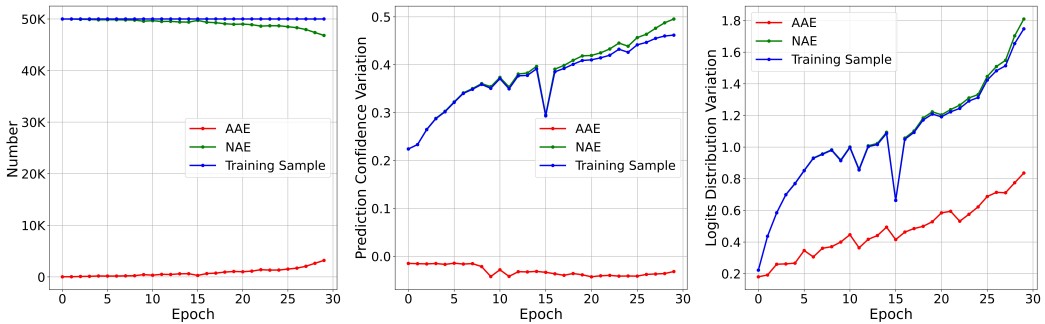

Figure 5. The number, the variation of prediction confidence and logits distribution (from left to right) for NAEs, AAEs and training samples in RS-AAER with 16/255 noise magnitude.

attempt to use the N-FGSM augmentation on Grad Align. However, we could not find suitable hyperparameter settings for Grad Align with the original step size. After reducing the step size to $\alpha = 1.0 \cdot \epsilon$ (consistent with N-AAER), the performance of Grad Align is similar to that reported in Table 2 (left). Besides, AAEs are not a unique phenomenon in SSAT, we also found them in the multi-step AT, indicating the existence of non-smooth points in their classifiers. However, we could not find competitive hyperparameter settings to enhance the robustness of multi-step AT using AAER.

**CIFAR100 Results.** We also conduct experiments on the CIFAR100 dataset, and the results are summarized in Table 2 (right). It is worth noting that CIFAR100 is more challenging as the number of classes/training images per class is ten times larger/smaller than that of CIFAR10. However, our proposed methods demonstrate their effectiveness in preventing CO and improving robust accuracy on CIFAR-100 as well. These results further validate that AAER is capable of reliably preventing CO and effectively improving robustness across different datasets.

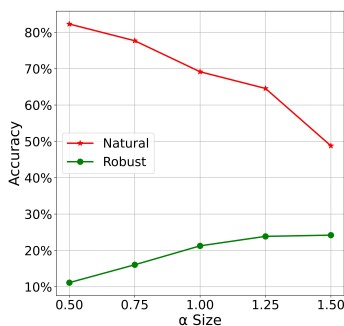

Figure 6. The role of $\alpha$.

### 4.3 Ablation Studies

In this part, we investigate the impacts of RS-AAER components with 16/255 noise magnitude using PreactResNet-18 on CIFAR10 under $L_\infty$ threat model.

**Optimization Objectives.** To validate the effectiveness of our proposed method, we illustrate the variation in test accuracy and three optimization objectives during the training process. Figure 1 demonstrates that our approach can successfully prevent CO and continuously improves robustness throughout the training. Additionally, from Figure 5, we can observe that the number, the variation of prediction confidence and logits distribution for AAEs are effectively constrained during the training. Specifically, when CO occurs in RS-FGSM (9th epoch), these three optimization objectives in RS-AAER are 29, 14, and 24 times smaller than it, respectively.

**The Role of $\alpha$.** From Figure 6, we can observe that enlarging the $\alpha$ size will lead to an increase in the robust accuracy, and correspondingly a decrease in the natural accuracy. In light of this trade-off, we follow the baseline setting and do not adjust the $\alpha$ size.

**The impact of Regularization Term.** We further investigate the interaction between the three parts of our regularization term as shown in Table 3. We find that neither part ($i$) number nor part ($ii$) prediction confidence can independently prevent CO, and only using part

Table 3. The impact of regularization term.

| ($i$) | ($ii$) | ($iii$) | Natural Acc (%) | Robust Acc (%) |
|-------|--------|---------|-----------------|----------------|
| ✓ | | | 75.14 | 0.00 |
| | ✓ | | 77.22 | 0.00 |
| | | ✓ | 13.61 | 9.77 |
| ✓ | ✓ | | 76.19 | 0.00 |
| ✓ | | ✓ | 56.65 | 23.29 |
| | ✓ | ✓ | 16.68 | 12.56 |
| ✓ | ✓ | ✓ | 64.56 | 23.87 |

Table 4. CIFAR10 training time on a single NVIDIA RTX 4090 GPU using PreactResNet-18. The results are averaged over 30 epochs.

| Method | FreeAT | ZeroGrad | MultiGrad | Grad Align | RS/N-FGSM | RS/N-AAER | PGD-2 | PGD-10 |
|---|---|---|---|---|---|---|---|---|
| Training Time (S) | 43.8 | 11.0 | 21.7 | 36.1 | 11.0 | 11.2 | 16.4 | 59.1 |

(*iii*) logits distribution can partially mitigate CO. However, solely relying on part (*iii*) cannot accurately reflect the degree of classifier distortion as it lacks a comprehensive measure, resulting in poor performance. Additionally, part (*i*) also plays an important role in performance, without it, both natural and robust accuracy significantly drop. Meanwhile, part (*ii*) contributes to the stability and natural accuracy of the method. Therefore, to effectively and stably eliminate CO, all parts of the regularization term are necessary and critical. Further ablation studies on other regularization methods and $\lambda$ selection can be found in Appendix B.

### 4.4 Time Complexity Study

Efficiency is a key advantage of SSAT over multi-step AT, as it can be readily scaled to large networks and datasets. Consequently, the computational overhead plays an important role in the SSAT overall performance. In Table 4, we present a time complexity comparison among various SSAT methods. It can be seen that AAER only imposes a minor training cost of 0.2 seconds, representing a mere 1.8% increase compared to FGSM. In contrast, Grad Align and PGD-10 are 3.2 and 5.3 times slower than our method.

## 5   Conclusion

In this paper, we find that the abnormal adversarial examples exhibit anomalous behaviour, i.e. they are further to the decision boundaries after adding perturbations generated by the inner maximization process. We empirically show the abnormal adversarial examples are closely related to the classifier distortion and catastrophic overfitting, by analyzing their number and outputs variation during the training process. Motivated by this, we propose a novel and effective method, *Abnormal Adversarial Examples Regularization* (AAER), through a regularizer to eliminate catastrophic overfitting by suppressing the generation of abnormal adversarial examples. Our approach can successfully resolve the catastrophic overfitting with different noise magnitudes and further boost adversarial robustness with negligible additional computational overhead.

### Acknowledgments and Disclosure of Funding

Tongliang Liu is partially supported by the following Australian Research Council projects: FT220100318, DP220102121, LP220100527, LP220200949, and IC190100031.

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

## A  Vanilla-AAER

To further validate the effectiveness of our method, we implement Vanilla-AAER to prevent CO. The Vanilla-AAER method follows the settings of Vanilla-FGSM [11], which does not use random initialization and sets the step size as $\alpha = 1.0 \cdot \epsilon$. The Vanilla-AAER hyperparameters setting is shown in Table 5.

Table 5. The hyperparameters setting for different noise magnitudes. The top number is $\lambda_2$ while the bottom number is $\lambda_3$. The $\lambda_1$ is fixed as 1.0 in all settings.

| Vanilla-AAER | 8/255 | 12/255 | 16/255 | 32/255 |
|:---:|:---:|:---:|:---:|:---:|
| $\lambda_2$ | 5.5 | 6.5 | 7.0 | 4.8 |
| $\lambda_3$ | 2.0 | 3.5 | 3.5 | 0.7 |

Table 6. CIFAR10: Accuracy of Vanilla-FGSM and Vanilla-AAER with different noise magnitude using PreActResNet-18 under $L_\infty$ threat model. The top number is the natural accuracy (%), while the bottom number is the PGD-50-10 accuracy (%).

| noise magnitude | 8/255 | 12/255 | 16/255 | 32/255 |
|:---:|:---:|:---:|:---:|:---:|
| Vanilla-FGSM | $84.16 \pm 4.68$ 
 $0.00 \pm 0.00$ | $79.86 \pm 2.05$ 
 $0.00 \pm 0.00$ | $72.51 \pm 3.79$ 
 $0.00 \pm 0.00$ | $64.29 \pm 3.83$ 
 $0.00 \pm 0.00$ |
| Vanilla-AAER | $80.45 \pm 0.25$ 
 $\mathbf{46.66 \pm 0.74}$ | $64.97 \pm 3.14$ 
 $\mathbf{32.44 \pm 1.18}$ | $51.92 \pm 2.90$ 
 $\mathbf{24.12 \pm 0.76}$ | $18.78 \pm 2.45$ 
 $\mathbf{12.19 \pm 0.40}$ |

Based on the results presented in Table 6, we can observe that Vanilla-AAER achieves comparable or even superior robustness compared to RS-AAER. This outcome may be attributed to the fact that Vanilla-FGSM has a higher expectation of adversarial perturbation, as demonstrated in prior work [1]. However, the absence of random initialization in Vanilla-AAER may reduce the diversity of adversarial examples, potentially impacting the natural accuracy of the model. Nonetheless, the most significant finding is that Vanilla-AAER effectively eliminates CO across various noise magnitudes, which cannot be accomplished by Vanilla-FGSM.

## B  Impacts of Regularization Term

To showcase the distinct effectiveness of our method in eliminating CO, we conducted a comparison with other regularization methods used in multi-step AT, such as TRADES [46] and ALP [19]. In order to ensure a fair comparison, we set the iteration times for TRADES and ALP as 1, and the step size as $\alpha = 1.25 \cdot \epsilon$ and $\alpha = 1.0 \cdot \epsilon$ for superimposing RS-FGSM and N-FGSM, respectively.

From Table 7, we can observe that TRADES and ALP methods may improve adversarial robustness when CO is not present in the baseline methods. However, these methods are not effective in eliminating CO. As demonstrated by the RS-TRADES and RS-ALP methods, CO still occurs with larger noise magnitudes, similar to RS-FGSM. Furthermore, these methods can even harm the baseline method, as they break the N-FGSM robustness with 32/255 noise magnitude. Therefore, we conclude that the TRADES and ALP methods are not suitable for eliminating CO. Additionally, other multi-step and robust overfitting methods also prove ineffective against CO. Hence, CO has been identified as an independent phenomenon requiring distinct solutions. In contrast to these regularization methods, AAER explicitly reflects and prevents distortion of the classifier from the perspective of AAEs, which is the key factor enabling AAER to effectively eliminate CO.

We further investigate the impact of hyperparameters $\lambda_1$, $\lambda_2$, and $\lambda_3$ on the performance of AAER. Figure 7 (left) shows the effect of $\lambda_1$ on the performance. It can be observed that when $\lambda_1$ is small, AAER is unable to effectively suppress CO, and increasing $\lambda_1$ improves both natural and robust accuracy. However, when $\lambda_1$ increases from 1.0 to 2.0, the robust accuracy remains unchanged while the natural accuracy decreases. Therefore, we choose $\lambda_1 = 1$ to balance both natural and robust

Table 7. CIFAR10: Accuracy of TRADES and ALP methods with different noise magnitude using PreActResNet-18 under $L_\infty$ threat model. The top number is the natural accuracy (%), while the bottom number is the PGD-50-10 accuracy (%).

| noise magnitude | 8/255 | 12/255 | 16/255 | 32/255 |
|---|---|---|---|---|
| RS-TRADES ($\beta = 1.0$) | 89.03 35.56 | 91.41 0.86 | 92.11 0.00 | 90.95 0.00 |
| RS-TRADES ($\beta = 6.0$) | 90.72 11.29 | 92.19 0.03 | 91.50 0.01 | 88.69 0.00 |
| RS-ALP (logit pairing weight=0.5) | 86.75 43.96 | 92.18 0.04 | 91.14 0.00 | 81.03 0.00 |
| RS-ALP (logit pairing weight=1.0) | 85.15 46.59 | 92.14 0.02 | 90.48 0.00 | 81.03 0.00 |
| N-TRADES ($\beta = 1.0$) | 86.82 41.24 | 81.12 16.40 | 85.62 0.12 | 81.81 0.00 |
| N-TRADES ($\beta = 6.0$) | 83.23 49.56 | 74.77 35.98 | 68.16 26.13 | 85.97 0.24 |
| N-ALP (logit pairing weight=0.5) | 84.63 46.03 | 81.57 27.85 | 72.05 0.32 | 79.41 0.00 |
| N-ALP (logit pairing weight=1.0) | 82.60 48.32 | 76.84 33.36 | 69.32 23.46 | 82.98 0.00 |

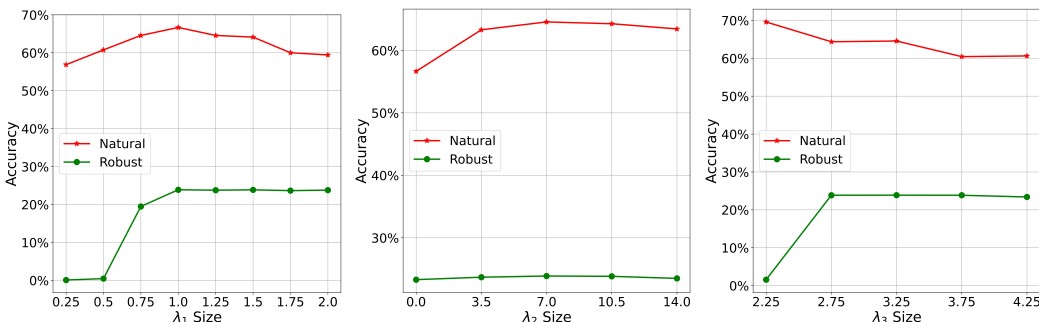

Figure 7. The role of $\lambda_1$ $\lambda_2$ and $\lambda_3$ under 16/255 noise magnitude (from left to right).

accuracy. Figure 7 (middle) demonstrates the impact of $\lambda_2$. It can be observed that when $\lambda_2$ is small, increasing $\lambda_2$ is beneficial for both natural and robust accuracy. However, when $\lambda_2$ increases from 7 to 14, the robust accuracy becomes flat while the natural accuracy decreases. Therefore, we select $\lambda_2 = 7$ considering both natural and robust accuracy. Figure 7 (right) shows the effect of $\lambda_3$. It can be observed that when $\lambda_3$ is small (2.25), the model experiences CO. Increasing $\lambda_3$ reduces the natural accuracy. From the variation of $\lambda_3$ between 2.25 and 4.75, we choose $\lambda_3 = 3.25$ to balance the elimination of CO with natural and robustness performance.

## C   Evaluation Based on Auto Attack

Auto Attack [5] is regarded as the most reliable robustness evaluation to date. It is an ensemble of complementary attacks, consisting of three white-box attacks (APGD-CE, APGD-DLR, and FAB) and a black-box attack (Square Attack). In order to avoid the pseudo-robustness brought by gradient masking or gradient obfuscation, we report the Auto Attack results on Cifar10/100 in Table 8 and Table 9.

In Table 8 and Table 9, we observe that our method, AAER, consistently improves adversarial robustness under Auto Attack on both the CIFAR-10 and CIFAR-100 datasets. It demonstrates that

AAER is effective in preventing CO and enhancing robustness under various adversarial attacks. The results validate the robustness and comprehensiveness of our proposed method.

Table 8. CIFAR10: Accuracy of different methods and different noise magnitudes using PreactResNet-18 under $L_\infty$ threat model. We only report the robust accuracy (%) under Auto Attack while the natural accuracy is same as Table 2. The results are averaged over 3 random seeds and reported with the standard deviation.

| noise magnitude | 8/255 | 12/255 | 16/255 | 32/255 |
|---|---|---|---|---|
| FreeAT | 40.23 ± 0.33 | 28.04 ± 0.73 | 0.00 ± 0.00 | 0.00 ± 0.00 |
| ZeroGrad | 43.48 | - | - | - |
| MulitGrad | 44.39 | - | - | - |
| Grad Align | 44.82 ± 0.09 | 30.05 ± 0.17 | 19.60 ± 0.47 | 7.89 ± 2.62 |
| RS-FGSM | 43.17 ± 0.34 | 0.00 ± 0.00 | 0.00 ± 0.00 | 0.00 ± 0.00 |
| N-FGSM | 44.43 ± 0.24 | 30.32 ± 0.08 | 19.06 ± 1.81 | 6.78 ± 0.75 |
| RS-AAER | 43.22 ± 0.20 | 26.75 ± 0.21 | 17.03 ± 0.51 | 5.37 ± 0.67 |
| N-AAER | **44.79 ± 0.23** | **30.76 ± 0.17** | **20.18 ± 0.15** | **8.46 ± 0.74** |
| PGD-2 | 42.97 ± 0.65 | 28.63 ± 0.38 | 18.52 ± 0.55 | 3.77 ± 0.02 |
| PGD-10 (20) | **46.95 ± 0.54** | **33.30 ± 0.20** | **22.29 ± 0.27** | **11.48 ± 0.43** |

Table 9. CIFAR100: Accuracy of different methods and different noise magnitudes using PreactResNet-18 under $L_\infty$ threat model. We only report the robust accuracy (%) under Auto Attack while the natural accuracy is same as Table 2. The results are averaged over 3 random seeds and reported with the standard deviation.

| noise magnitude | 8/255 | 12/255 | 16/255 | 32/255 |
|---|---|---|---|---|
| FreeAT | 18.28 ± 0.20 | 12.37 ± 0.14 | 0.00 ± 0.00 | 0.00 ± 0.00 |
| ZeroGrad | 21.15 | - | - | - |
| MulitGrad | 21.62 | - | - | - |
| Grad Align | 21.87 ± 0.13 | 13.78 ± 0.11 | 9.64 ± 0.12 | 1.76 ± 0.70 |
| RS-FGSM | 7.98 ± 11.91 | 0.00 ± 0.00 | 0.00 ± 0.00 | 0.00 ± 0.00 |
| N-FGSM | 22.68 ± 0.25 | 14.57 ± 0.09 | 10.30 ± 0.14 | 0.00 ± 0.00 |
| RS-AAER | 21.41 ±0.01 | 12.31± 0.28 | 8.56 ±0.02 | 2.93 ± 0.17 |
| N-AAER | **22.93 ± 0.10** | **14.73 ± 0.24** | **10.35 ± 0.11** | **3.46 ± 0.14** |
| PGD-2 | 22.52 ± 0.14 | 13.69 ± 0.02 | 9.56 ± 0.07 | 1.76 ± 0.22 |
| PGD-10 (20) | **23.78 ± 0.08** | **15.61 ± 0.09** | **10.93 ± 0.05** | **4.13 ± 0.10** |

# D   Experiment with WideResNet Architecture

We also compare the performance of our method using WideResNet-34, which is more complex than PreActResNet. Since the baselines cannot adapt well to WideResNet-34, we also need to correspondingly adjust the hyperparameters. The $\lambda_1$ is fixed as 1.0 in all settings. For CIFAR10, we set RS-AAER $\lambda_2 = 4.0$ and $\lambda_3 = 2.0$, N-AAER $\lambda_2 = 2.5$ and $\lambda_3 = 0.6$. For CIFAR100, we set RS-AAER $\lambda_2 = 2.5$ and $\lambda_3 = 1.0$, N-AAER $\lambda_2 = 1.0$ and $\lambda_3 = 0.2$. We report the results on Cifar10/100 in Table 10 and Table 11.

Table 10. CIFAR10: Accuracy of different methods with 8/255 noise magnitude using WideResNet-34 under $L_\infty$ threat model. The results are averaged over 3 random seeds and reported with the standard deviation.

| method | RS-FGSM | N-FGSM | RS-AAER | N-AAER | PGD-2 | PGD-10 |
|---|---|---|---|---|---|---|
| natural accuracy (%) | 84.41 ± 0.45 | 84.67 ± 0.32 | 87.39 ± 0.14 | 84.47 ± 0.23 | 88.68 ± 0.14 | 85.53 ± 0.22 |
| robust accuracy (%) | 0.00 ± 0.00 | 49.72 ± 0.25 | 47.58 ± 0.42 | **50.07 ± 0.53** | 47.32 ± 0.50 | **53.70 ± 0.53** |
| training time (S) | 98.2 | | 98.6 | | 147.1 | 536.2 |

Table 11. CIFAR100: Accuracy of different methods with 8/255 noise magnitude using WideResNet-34 under $L_\infty$ threat model. The results are averaged over 3 random seeds and reported with the standard deviation.

| method | RS-FGSM | N-FGSM | RS-AAER | N-AAER | PGD-2 | PGD-10 |
|---|---|---|---|---|---|---|
| natural accuracy (%) | 55.04 ± 1.24 | 59.02 ± 0.63 | 59.81 ± 0.38 | 57.76 ± 0.36 | 64.64 ± 0.27 | 60.34 ± 0.34 |
| robust accuracy (%) | 0.00 ± 0.00 | 28.49 ± 0.54 | 26.88 ± 0.30 | **29.09 ± 0.66** | 26.47 ± 0.10 | **30.02 ± 0.09** |

In Table 10 and Table 11, we observe that when using the WideResNet-34 architecture, RS-FGSM suffers from CO with a noise magnitude of 8/255, which is different from the results obtained with the PreActResNet-18 architecture. However, our method, AAER, can successfully prevent CO and achieve high robustness even with complex network architectures. This demonstrates the reliability of AAER in preventing CO and improving robustness across different network architectures. It is worth noting that complex networks can better reflect the efficiency of our method in terms of training time, while our method can achieve comparable robustness to multi-step AT.

## E  Settings and Results on SVHN, Tiny-ImageNet and Imagenet-100

**SVHN Settings and Results.** For experiments on SVHN, we use the cyclical learning rate schedule with 15 epochs that reaches its maximum learning rate (0.05) when 40% (6) epochs are passed. In the meantime, we uniformly increase the step size between 0 and 5 epochs, which follow the settings of [6]. We show the hyperparameters setting on SVHN in Table 12. In Table 13, we present the performance of AAER on the SVHN dataset, along with the results of the competing baseline taken from [6]. It is evident that our method can successfully prevent CO and improve robust accuracy across different noise magnitudes. This demonstrates the effectiveness of AAER in enhancing the robustness of models trained on the SVHN dataset.

Table 12. SVHN: The hyperparameters setting for different noise magnitudes. The top number is $\lambda_2$ while the bottom number is $\lambda_3$. The $\lambda_1$ is fixed as 1.0 in all settings.

| SVHN | 4/255 | 8/255 | 12/255 |
|---|---|---|---|
| RS-AAER | 0.5 
 1.25 | 0.6 
 0.85 | 0.45 
 0.55 |
| N-AAER | 0.75 
 0.25 | 1.0 
 1.0 | 1.0 
 0.75 |

**Tiny-ImageNet Settings and Results.** We also scale our method to a medium-sized dataset Tiny-ImageNet to showcase its effectiveness. We utilized the cyclical learning rate schedule with 30 epochs, reaching the maximum learning rate of 0.2 at the midpoint of 15 epochs. For RS-AAER, we set $\lambda_2 = 0.75$ and $\lambda_3 = 0.15$, while for N-AAER, we set $\lambda_2 = 0.25$ and $\lambda_3 = 0.05$. The value of $\lambda_1$ remained fixed at 1.0 in all settings. Table 14 presents the performance of AAER on the Tiny-ImageNet dataset. We can observe that our method effectively prevents CO and improves robust accuracy in this medium-scale dataset.

Table 13. SVHN: Accuracy of different methods and different noise magnitudes using PreActResNet-18 under $L_\infty$ threat model. The baseline results are taken from [6]. The top number is the natural accuracy (%), while the bottom number is the PGD-50-10 accuracy (%). The results are averaged over 3 random seeds and reported with the standard deviation.

| noise magnitude | 4/255 | 8/255 | 12/255 |
|---|---|---|---|
| FreeAT | 93.66 ± 0.12 | 91.29 ± 4.07 | 92.36 ± 1.00 |
|  | 71.61 ± 0.75 | 0.01 ± 0.00 | 0.00 ± 0.00 |
| ZeroGrad | 94.81 ± 0.16 | 92.42 ± 1.29 | 88.09 ± 0.40 |
|  | 71.59 ± 0.22 | 35.93 ± 2.73 | 14.14 ± 0.32 |
| MultiGrad | 94.71 ± 0.17 | 94.86 ± 0.97 | 94.48 ± 0.19 |
|  | 71.98 ± 0.26 | 11.49 ± 16.19 | 0.00 ± 0.00 |
| Grad Align | 94.56 ± 0.21 | 90.1 ± 0.34 | 84.01 ± 0.46 |
|  | 72.12 ± 0.19 | 43.85 ± 0.14 | 23.62 ± 0.41 |
| RS-FGSM | 95.09 ± 0.09 | 94.46 ± 0.16 | 92.74 ± 0.5 |
|  | 71.28 ± 0.40 | 0.00 ± 0.00 | 0.00 ± 0.00 |
| N-FGSM | 94.54 ± 0.15 | 89.56 ± 0.49 | 81.48 ± 1.64 |
|  | 72.53 ± 0.19 | 45.63 ± 0.11 | 26.13 ± 0.81 |
| RS-AAER | 94.99 ± 0.70 | 90.11 ± 0.85 | 83.50 ± 4.13 |
|  | 71.97 ± 0.88 | 41.75 ± 0.55 | 22.84 ± 0.51 |
| N-AAER | 94.35 ± 0.26 | 89.26 ± 0.57 | 82.76 ± 0.84 |
|  | **73.10 ± 0.23** | **46.98 ± 0.25** | **26.87 ± 1.51** |
| PGD-2 | 94.66 ± 0.10 | 94.63 ± 1.29 | 94.16 ± 0.54 |
|  | 73.29 ± 0.29 | 20.68 ± 18.56 | 0.02 ± 0.03 |
| PGD-10 | 94.37 ± 0.13 | 89.67 ± 0.34 | 80.08 ± 0.93 |
|  | **74.76 ± 0.19** | **53.95 ± 0.55** | **37.65 ± 0.53** |

Table 14. Tiny-ImageNet: Accuracy of different methods with 8/255 noise magnitude using PreActResNet-18 under $L_\infty$ threat model. The results are averaged over 3 random seeds and reported with the standard deviation.

| method | RS-FGSM | N-FGSM | RS-AAER | N-AAER | PGD-2 |
|---|---|---|---|---|---|
| natural accuracy (%) | 52.28 ± 2.64 | 48.16 ± 0.61 | 49.86 ± 0.39 | 47.93 ± 0.27 | 46.43 ± 0.35 |
| robust accuracy (%) | 0.00 ± 0.00 | 20.73 ± 0.40 | 19.66 ± 0.14 | **20.92 ± 0.01** | 20.72 ± 0.32 |

Table 15. ImageNet-100: Accuracy of different methods with 8/255 noise magnitude using PreActResNet-18 under $L_\infty$ threat model. The results are averaged over 3 random seeds and reported with the standard deviation.

| method | RS-FGSM | N-FGSM | RS-AAER | N-AAER |
|---|---|---|---|---|
| natural accuracy (%) | 27.10 ± 11.44 | 38.87 ± 0.17 | 32.28 ± 1.52 | 39.52 ± 0.42 |
| robust accuracy (%) | 0.00 ± 0.00 | 20.71 ± 0.74 | 14.22 ± 0.96 | **20.90 ± 0.34** |

**ImageNet-100 Settings and Results.** We have also extended our method to a large-sized dataset, ImageNet-100, to demonstrate its effectiveness. We utilized the cyclical learning rate schedule with 30 epochs, reaching the maximum learning rate of 0.2 at the midpoint of 15 epochs. For RS-AAER, we set $\lambda_2 = 3.0$ and $\lambda_3 = 2.5$, while for N-AAER, we set $\lambda_2 = 1.25$ and $\lambda_3 = 0.25$. The value of $\lambda_1$ remained fixed at 1.0 in all settings. From Table 15, it is evident that our method effectively prevents CO and enhances robust accuracy in this large-scale dataset. It needs to be highlighted that the results in the above table may not be optimal, which can be further improved by increasing training epochs or adjusting hyperparameters and learning rate. However, they do establish the effectiveness of our method in trustworthyly eliminating CO on a larger-scale dataset. The above outcomes underscore the scalability and effectiveness of AAER in fortifying the robustness of models trained.

# F   More Competing Baselines

We also compare the performance of our method with other SSAT methods, including GAT [33], NuAT [34], PGI-FGSM [17], SDI-FGSM [18] and Kim [21]. The results for GAT, NuAT and Kim are directly taken from [6], while the reported PGI-FGSM and SDI-FGSM results are based on the official code after searching the hyperparameters across different noise magnitudes.

Table 16.  CIFAR10:  Accuracy of different methods and different noise magnitudes using PreActResNet-18 under $L_\infty$ threat model. GAT, NuAT and Kim results are taken from [6]. The top number is the natural accuracy (%), while the bottom number is the PGD-50-10 accuracy (%). The results are averaged over 3 random seeds and reported with the standard deviation.

| noise magnitude | 8/255 | 12/255 | 16/255 |
|---|---|---|---|
| GAT | $76.75 \pm 0.38$ $50.98 \pm 0.12$ | $80.44 \pm 5.08$ $14.93 \pm 9.26$ | $82.17 \pm 2.47$ $1.25 \pm 0.51$ |
| NuAT | $73.22 \pm 0.34$ $50.10 \pm 0.33$ | $74.38 \pm 7.32$ $17.54 \pm 8.82$ | $80.1 \pm 1.08$ $3.29 \pm 0.87$ |
| PGI-FGSM | $77.94 \pm 0.26$ $\mathbf{52.86 \pm 0.34}$ | $83.82 \pm 0.86$ $5.19 \pm 0.59$ | $83.42 \pm 0.24$ $4.16 \pm 0.31$ |
| SDI-FGSM | $79.28 \pm 0.08$ $49.26 \pm 0.10$ | $70.07 \pm 0.84$ $\mathbf{36.56 \pm 1.34}$ | $81.09 \pm 0.13$ $0.05 \pm 0.01$ |
| Kim | $89.02 \pm 0.10$ $33.01 \pm 0.09$ | $88.35 \pm 0.31$ $13.11 \pm 0.63$ | $90.45 \pm 0.08$ $1.88 \pm 0.05$ |
| RS-AAER | $83.83 \pm 0.27$ $46.14 \pm 0.02$ | $74.40 \pm 0.79$ $32.17 \pm 0.16$ | $64.56 \pm 1.45$ $23.87 \pm 0.36$ |
| N-AAER | $80.56 \pm 0.35$ $48.31 \pm 0.23$ | $71.15 \pm 0.18$ $\mathbf{36.52 \pm 0.10}$ | $61.84 \pm 0.43$ $\mathbf{28.20 \pm 0.71}$ |

In Table 16, we can observe that GAT, NuAT, PGI-FGSM and SDI-FGSM demonstrate superior performance under 8/255 noise magnitude. However, it becomes apparent that these methods still suffer from CO with strong adversaries, resulting in nearly zero robustness under 16/255 noise magnitude, let alone the more challenging 32/255. In contrast, our proposed method exhibits consistent effects across different settings with negligible computational overhead, demonstrating its trustworthy effectiveness in preventing CO. We would like to emphasize that while achieving excellent performance under 8/255 noise magnitude is certainly gratifying, but can reliable defence against CO is more critical to a successful SSAT method.

# G   Long Training Schedule

We also compare the performance of our method using the standard robust overfitting training schedule. This training schedule consists of 200 epochs with an initial learning rate of 0.1. The learning rate is divided by 10 at the 100th and 150th epochs, respectively. During the long training

schedule, we employ a warm-up strategy in the first 20 epochs, which uniformly rises the strength of AAER from 0% to 100%.

Table 17. CIFAR10: Accuracy of long training schedule with 8/255 noise magnitude using PreActResNet-18 under $L_\infty$ threat model. The results are averaged over 3 random seeds and reported with the standard deviation.

| method | RS-FGSM | N-FGSM | RS-AAER | N-AAER |
|---|---|---|---|---|
| natural accuracy (%) | 91.21 ± 0.26 | 83.25 ± 0.04 | 85.69 ± 0.20 | 83.23 ± 0.25 |
| robust accuracy (%) | 0.00 ± 0.00 | 36.98 ± 0.34 | 36.05 ± 0.17 | **37.38 ± 0.16** |

In Table 17, we observe that our method, AAER, consistently improves adversarial robustness under the long training schedule, underscoring its consistently reliable and effective performance in preventing CO.

