# OpenReview forum: "Eliminating Catastrophic Overfitting Via Abnormal Adversarial Examples Regularization"
_NeurIPS.cc/2023/Conference — NeurIPS 2023 poster_

### Official Review · Reviewer_6vch · 2023-07-06

**Soundness:** 3 good
**Presentation:** 3 good
**Contribution:** 3 good
**Rating:** 4
**Confidence:** 4

**Summary:**

The authors observe that the loss of some adversarial examples generated by the inner maximization process during training decreases, which they call abnormal adversarial examples (AAEs). Based on this, they discover a relationship between AAEs and catastrophic overfitting. Therefore, they propose an abnormal adversarial examples regularization to hinder the classifier from becoming distorted.

**Strengths:**

The finding and definition of abnormal adversarial examples are interesting. The corresponding analysis is comprehensive.
The proposed method can effectively alleviate catastrophic overfitting when the perturbation radius increases.


**Weaknesses:**

The experiments were not sufficient to comprehensively evaluate the proposed method. Experiments on a larger dataset (ImageNet-100, ImageNet) should be conducted. In addition, GAT (NeurIPS 2020) and FGSM-PGI (ECCV 2022) could be added as baselines.
30 epochs are insufficient. Experiments with more epochs should be involved to show that the proposed method can eliminate catastrophic overfitting.


**Questions:**

It seems that this method needs to find out AAEs and NAEs for each minibatch, how much extra training time does this require?

**Limitations:**

See above.

---

> ### Author Rebuttal · Authors · 2023-08-10
>
> We sincerely appreciate your time and effort in reviewing our manuscript. Following thorough consideration, please find the responses to your comments below.
>
> >**Q1: Insufficient experiments**
>
> **A1:** We acknowledge your concerns and are committed to addressing them comprehensively. In Appendix E, we have already reported the results of our method's effectiveness on the medium-sized dataset Tiny-imagenet. To further address this concern, we report the ImageNet-100 results below. For Imagenet-100, we utilized the cyclical learning rate schedule with 30 epochs, reaching the maximum learning rate of 0.2 at the midpoint of 15 epochs. For RS-AAER, we set $\lambda_{2} = 0.75$ and $\lambda_{3} = 0.15$, while for N-AAER, we set $\lambda_{2} = 0.25$ and $\lambda_{3} = 0.02$. The value of $\lambda_{1}$ remained fixed at 1.0 in all settings. We report both natural and **robust** accuracy of the final model on the ImageNet-100 dataset under 8/255 noise magnitude.
>
> |Method |RS-FGSM|N-FGSM|RS-AAER|N-AAER|
> |:-----|:----:|:----:|:----:|:----:|
> |Natural Acc (%)|38.78|39.20|38.66|38.64|
> |Robust Acc (%)|**0.00**|**0.00**|**20.34**|**20.58**|
>
> It needs to highlight that the results in the above table may not be optimal, which can be further improved by increasing training epochs or adjusting hyperparameters and learning rate. But, they do establish the effectiveness of our method in trustworthy eliminating CO on a larger-scale dataset.
>
> >**Q2: More competing baselines**
>
> **A2:** Please refer to **Q/A1** in general response.
>
> >**Q3: Long training schedule**
>
> **A3:** We utilize the standard robust overfitting training schedule to evaluate the effectiveness of our method. This training schedule consists of 200 epochs with an initial learning rate of 0.1. The learning rate is divided by 10 at the 100-th and 150-th epochs, respectively. During the long training schedule, we employ a warm-up strategy in the first 20 epochs, which uniformly rises the strength of AAER from 0% to 100%. We report both natural and **robust** accuracy of the final model on the CIFAR10 dataset under 8/255 noise magnitude.
>
> |Method|RS-FGSM|N-FGSM|RS-AAER|N-AAER|
> |:-----|:----:|:----:|:----:|:----:|
> |Natural Acc (%)|91.21|83.25|85.69|83.23|
> |Robust Acc (%)|**0.13**|**36.98**|**36.05**|**37.38**|
>
> Based on the above table, our proposed method demonstrates robustness with long training schedules, underscoring its consistently reliable and effective performance in preventing CO.
>
> >**Q4: Computational overhead**
>
> **A4:** Please refer to **Q/A2** in general response.
>
> If you have any further questions or concerns, please do not hesitate to contact us. We are willing and available to provide any additional information or clarification you may need.

---

> ### Author Response · Authors · 2023-08-17
> **Response to Reviewer 6vch**
>
> We hope this message finds you well. We express our sincere appreciation for the time and effort you have dedicated to reviewing our manuscript. As a reminder, we have submitted a rebuttal to address your concerns. In the rebuttal, we validated the effectiveness of our method using large-sized datasets and a long training schedule. Additionally, we reported the training time to demonstrate its efficiency. We understand that the discussion period can be tight, and your prompt response would enable us in addressing any additional concerns. We are looking forward to hearing from you soon.
>
> Best regards,
>
> The Authors

---

### Official Review · Reviewer_EF7n · 2023-07-06

**Soundness:** 3 good
**Presentation:** 4 excellent
**Contribution:** 3 good
**Rating:** 4
**Confidence:** 5

**Summary:**

This work finds that abnormal adversarial examples(AAE) are generated during single-step adversarial training. And AAE has a deep relationship with catastrophic overfitting. According to the observation, authors propose a new regularization method AAER to regularize the numbers and output variations of AAER. Results show that their method can improve the performances of existed SSAT methods.

**Strengths:**

1. Figures in this work express their ideas and observations clearly, such as Figure 2.
2. The method work for existed SOTA single-step adversarial training methods.
3. Results of diverse networks and datasets are provided.
4. Confidence results are provided to show the stability of methods.


**Weaknesses:**

1. Compared to N-FGSM, N-AAER does not have an obvious improvement and takes more time.
2. AAER can improve the clean performance sometimes. For example, RS-AAER improves the clean performance a lot, 8%, compared to RS-FGSM when the $\epsilon$ = 16/255. However, RS-AAER decreases the clean performance when the $\epsilon$ = 32/255. And for N-FGSM, AAER always decrease the clean performance. What's the role of AAER for clean performances?
3. In general, the high performance of AAER is build on previous successful training methods. N-AAER is the best one because N-FGSM is the best one. It's difficult to say it's the AAER make the success as the improvement brought by AAER is not that obvious.

**Questions:**

1. Does the method AAER improve the performance of GradAlign?
2. Will the extra training cost brought by AAER increase along with the dataset size? If not, what is the training cost related to?
3. Could the AAER improve the performance of PGD-AT?


**Limitations:**

This work is about the safety in deep learning.

---

> ### Author Rebuttal · Authors · 2023-08-10
>
> We sincerely appreciate your time and effort in reviewing our manuscript. Following thorough consideration, please find the responses to your comments below.
>
> >**Q1 & 3: Primary objective**
>
> **A1 & 3:** Our primary objective is to eliminate CO in SSAT methods. Therefore, our method is specifically designed to address the CO, which will demonstrate significant performance improvements under CO scenarios, such as Vanilla-AAER (under 8, 12, 16, 32/255 noise magnitude), RS-AAER (under 12, 16, 32/255 noise magnitude), and N-AAER (under 32/255 noise magnitude). Besides, without the CO scenario, our approach still has complementary robustness and consistently outperforms the baselines. This clearly validates our motivation and demonstrates the effectiveness of our method.
>
> >**Q2: Trade-off between robustness and clean accuracy**
>
> **A2:** In AT, the model generally exhibits a trade-off between robustness and clean accuracy, as mentioned in [1]. For instance, N-AAER achieves higher robustness than N-FGSM at the expense of a slight reduction in natural accuracy. However, this trade-off does not suitable for CO models (still suitable for SSAT models without CO). Once CO occurs, the model will become fully distorted and collapse, leading to a plummeting drop in robust accuracy and a highly random and unpredictable clean accuracy (indicated by the high standard deviation reported in Tab.1). The collapse of the decision boundary significantly impacts the model's overall performance, rendering the traditional trade-off and clean accuracy irrelevant within CO phenomenon.
>
> >**Q4: AAER with Grad Align**
>
> **A4:** Both goals of AAER and Grad Align can be considered as implicitly smoothing the loss landscape. Therefore, when combining AAER and Grad Align, although robustness does not decrease, we also did not observe a significant enhancement in robustness. However, it is noteworthy that our proposed AAER method performs comparably or even better than Grad Align, while requiring only one-third of the training cost which is very crucial in SSAT.
>
> >**Q5: Training Cost**
>
> **A5:** The AAER method does not involve any extra generation, forward, or backward propagation processes, all additional processing is solely based on the original FGSM intermediate variables. Moreover, all AAER processing steps, including dividing AAEs, and calculating CE loss and L2 norm, can be efficiently executed using GPU parallel computing. As a result, the additional time overhead of AAER is directly proportional to the original FGSM training time, and this ratio remains constant regardless of changes in the dataset or model structure. **Additionally, we are delighted to report that we have observed a minimal 1.8% increase in RS/N-AAER time overhead compared to RS/N-FGSM. This observation was made using a single NVIDIA RTX 4090 GPU with PyTorch 2.0, and all experiments without half-precision float.**
>
> |Method|FreeAT|ZeroGrad|MultiGrad|Grad Align|RS\N-FGSM|RS\N-AAER|PGD-2|PGD-10|
> |:-----|:----:|:----:|:----:|:----:|:----:|:----:|:----:|:----:|
> |Training Time (S)|43.8|11.0|21.7|36.1|11.0|11.2|16.4|59.1|
> |Ratio (%)|391.07|98.21|193.75|322.32|98.21|100%|146.42|527.67|
>
> **NOTE: AAER as a baseline for training time ratio.**
>
> >**Q6: AAER with PGD-AT**
>
> **A6:** AAE is not a unique phenomenon in SSAT, but also exists in iterative-step AT methods, indicating the existence of non-smooth points in their classifiers. However, we could not find competitive hyperparameter settings to significantly enhance the robustness of multi-step AT using AAER.
>
> If you have any further questions or concerns, please do not hesitate to contact us. We are willing and available to provide any additional information or clarification you may need.
>
> [1] Zhang et al. "Theoretically principled trade-off between robustness and accuracy." ICML 2019.

---

> ### Author Response · Authors · 2023-08-17
> **Response to Reviewer EF7n**
>
> We hope this message finds you well. We express our sincere appreciation for the time and effort you have dedicated to reviewing our manuscript. As a reminder, we have submitted a rebuttal to address your concerns. In the rebuttal, we emphasized our primary objective of effectively preventing CO and clarified that the trade-off is not suitable for the CO model. Moreover, we reported training time to demonstrate the efficiency of our method. We understand that the discussion period can be tight, and your prompt response would enable us in addressing any additional concerns. We are looking forward to hearing from you soon.
>
> Best regards,
>
> The Authors

---

### Official Review · Reviewer_N8c9 · 2023-07-07

**Soundness:** 2 fair
**Presentation:** 3 good
**Contribution:** 2 fair
**Rating:** 4
**Confidence:** 4

**Summary:**

The paper deals with the mitigation of catastrophic overfitting in FGSM adversarial training. The paper first presents the properties of adversarial samples before and after catastrophic overfitting. Finally, the paper proposes a regularizer (Abnormal Adversarial Examples Regularization AAER) to mitigate catastrophic overfitting during FGSM AT. The proposed regularizer prevents the generation of abnormal adversarial samples and stabilizes the training process. The effectiveness of the single-step AT with the proposed regularizer is demonstrated on CIFAR 10/100, SVHN, and TinyImageNet datasets.

**Strengths:**

The paper presents important observations on adversarial samples before and after catastrophic overfitting (CO) (i.e., presence of pseudo-adversarial samples, population and properties of pseudo-adversarial samples before and after CO). Though not novel, these observations are presented from a CO perspective. The proposed regularizer harnesses these observations to mitigate CO.

**Weaknesses:**

1. The paper presents important observations on adversarial samples before and after catastrophic overfitting. However, it fails to explain the cause for these observations (why do models generate abnormal adversarial samples during single-step AT?)

2. The observations presented in this paper are not novel [1,2]. The paper presents the extreme case of gradient masking, i.e., the label leaking [1], where the accuracy of the model on adversarial samples (with large perturbation size) is greater than clean accuracy. Furthermore, not all pseudo-robust models generate abnormal adversarial samples.

3. Comparison with existing robust single step adversarial training is missing [3-6]

4. The paper fails to demonstrate the effectiveness of the proposed regularizer. The proposed regularizer is plugged into existing robust single-step adversarial training methods (these methods mitigate CO) to show its effectiveness. Most of the results are shown in this setting, and a marginal improvement in robustness is observed. The robustness of the model trained using the vanilla variant is sub-par compared to models trained using existing robust single adversarial training methods (compare table-1 and 4 (supplementary)).

[1] Kurakin et al. "Adversarial machine learning at scale." arXiv  2016.

[2] Tramer et al. “Ensemble adversarial training: attacks and defense” arXiv  2017

[3] Sriramanan et al. "Guided adversarial attack for evaluating and enhancing adversarial defenses" NeuRIPS  2020

[4] Sriramanan et al. "Towards efficient and effective adversarial training." NeuRIPS  2021

[5] Kim et al. "Understanding catastrophic overfitting in single-step adversarial training" AAAI 2021.

[6] Jia et al. "Boosting fast adversarial training with learnable adversarial initialization" IEEE Transactions on Image Processing,  2022.

**Questions:**

1. Address the weakness.

2. Provide experimental details for results presented in all the tables and figures

3. Can the authors explain the observation considering min-max optimization AT formulation?

4. Provide the plot of loss vs perturbation size of FGSM and PGD-10 attack for models trained using the proposed approach.

5. Sensitivity of the algorithm to hyper-parameters: $\lambda_1$=1, $\lambda_2$=optimal value $\lambda_3$=0 and $\lambda_1$=1, $\lambda_2$=0, $\lambda_3$=optimal value

6. Report training time for N-AAER and RS-AAER.

**Limitations:**

The limitation of the proposed approach is not discussed. Authors are suggested to present failure mode/cases for the proposed approach.

---

> ### Author Rebuttal · Authors · 2023-08-10
>
> We sincerely appreciate your time and effort in reviewing our manuscript. Following thorough consideration, please find the responses to your comments below.
>
> >**Q1: Why does the model generate AAEs?**
>
> **A1:** The presence of AAEs is correlated with classifier distortion. By analyzing AAEs, we have addressed two unclear questions in SSAT: (1) the gradual distortion process of the classifier, and (2) the underlying factors of rapid classifier distortion. Our study provides valuable insights for future research to explore the underlying causes of the model's spontaneous distortion.
>
> >**Q2: The novelty of our work**
>
> **A2:** While we have observed similar phenomena as in previous studies [1, 2], it's important to emphasize that our research is grounded in a completely different research background and perspective. Notably, the concepts of PGD and CO had not been introduced at that time. Specifically, in terms of the research object, [1, 2] only focused on the final model's performance. In our work, we delved into the dynamic interaction between AAEs and CO throughout the training process, and find the presence of AAEs before the occurrence of CO. This finding reveals that before CO occurs (the model remains robust), the classifier already exhibits slight distortions. Secondly, regarding the research subject, [1, 2] treated all adversarial examples as a consistent sample. In contrast, we divide them into two groups: AAEs and NAEs, based on their distinct properties. This categorization enables us to delve into their individual impacts on the SSAT training process, and find that directly optimizing AAEs will accelerate the CO process. Lastly, for methodology, [1, 2] is based on the perspective of gradient masking, which cannot effective defence against CO. In comparison, from the perspective of AAEs, our proposed method demonstrates trustworthy effectiveness in preventing CO. The above valuable findings and effective method distinguish our work from the [1, 2].
>
> >**Q3: More competing baselines**
>
> **A3:** Please refer to **Q/A1** in general response.
>
> >**Q4: Primary objective**
>
> **A4:** Our primary objective is to eliminate CO in SSAT methods. Therefore, our method is specifically designed to address the CO, which will demonstrate significant performance improvements under CO scenarios, such as Vanilla-AAER (under 8, 12, 16, 32/255 noise magnitude), RS-AAER (under 12, 16, 32/255 noise magnitude), and N-AAER (under 32/255 noise magnitude). Besides, without the CO scenario, our approach still has complementary robustness and consistently outperforms the baselines. This clearly validates our motivation and demonstrates the effectiveness of our method.
>
> >**Q5: Experimental details**
>
> **A5:** In Section 4.1 "Experiment Settings: Setup for Our Proposed Method," we have provided comprehensive details of our experimental setup, including the optimizer, learning rate schedule, and hyperparameter settings, which follow the classical setup for SSAT [3]. Additionally, for a specific table, picture, and dataset, you can find detailed setting information in their respective descriptions or appendices within the submission.
>
> >**Q6: Min-max perspective**
>
> **A6:** From a min-max perspective, the inner maximization process tries to generate the strongest adversarial examples by gradient ascent. However, due to the distortion of the classifier, the generated adversarial example might rely on the incorrect linear approximation, leading to the generation of AAEs. Subsequently, the model optimizes these AAEs through gradient descent minimization, further learning incorrect linear approximation, thereby exacerbating the distortion of the classifier. This interaction leads to a vicious circle between the AAEs and the decision boundary distortion, ultimately leading to the model rapidly manifesting as CO.
>
> >**Q7: Loss vs perturbation size**
>
> **A7:** We observed that the attack loss and successful rate (100% - test accuracy) of both the PGD and FGSM attacks on the AAER-trained method gradually increase with the perturbation size increase. We report both attack loss and successful rate of the final RS-AAER on the CIFAR10 dataset below.
>
> |Noise Magnitude|8/255|12/255|16/255|32/255|
> |:-|:-:|:-:|:-:|:-:|
> |FGSM Loss|1.21|1.54|1.77|2.14|
> |FGSM Successful Rate (%)|45.46|58.13|65.67|79.88|
> |PGD Loss|0.91|1.26|1.50|2.06|
> |PGD Successful Rate (%)|53.70|67.46|76.02|88.69|
>
> >**Q8: Sensitivity of the hyperparameters**
>
> **A8:** In Section 4.3 "Ablation Studies: The Role of Regularization Term. Fig. 5: (i, ii)", we observed that AAER cannot effectively mitigate CO by relying solely on $\lambda_1$ = 1 and $\lambda_2$ = optimal value. On the contrary, AAER that exclusively depends on $\lambda_1$ =1 and $\lambda_3$ =optimal value can successfully achieve robustness against adversarial attacks, as shown in Fig. 5: (i, iii). However, the absence of $\lambda_2$ leads to an incomprehensive assessment of AAEs, resulting in lower natural accuracy. Therefore, to effectively and stably eliminate CO, all components of the regularization term are indispensable and critical.
>
> >**Q9: Computational overhead**
>
> **A9:** Please refer to **Q/A2** in general response.
>
> [1] Kurakin et al. "Adversarial machine learning at scale." arXiv 2016.
>
> [2] Tramer et al. “Ensemble adversarial training: attacks and defense.” arXiv 2017.
>
> [3] de Jorge Aranda et al. "Make some noise: Reliable and efficient single-step adversarial training." NeuRIPS 2022.

---

> ### Author Response · Authors · 2023-08-17
> **Response to Reviewer N8c9**
>
> We hope this message finds you well. We express our sincere appreciation for the time and effort you have dedicated to reviewing our manuscript. As a reminder, we have submitted a rebuttal to address your concerns. In the rebuttal, we emphasized the disparities in research background, problem, and objectives between our work and prior studies. Furthermore, we presented supplementary experiments to validate the effectiveness and reported training time to demonstrate the efficiency of our method. We understand that the discussion period can be tight, and your prompt response would enable us in addressing any additional concerns. We are looking forward to hearing from you soon.
>
> Best regards,
>
> The Authors

---

> > ### Comment · Reviewer_N8c9 · 2023-08-20
> >
> > Thanks for the response.
> > 1. Are AAEs present during the single-step adversarial training with small perturbation size (e.g., epsilon=2/255)?
> > 2. In the above table (Q7) noise vs. perturbation size, why is the loss on the FGSM (single-step) adversarial samples greater than PGD (multi-step) adversarial samples?

---

> > > ### Author Response · Authors · 2023-08-21
> > > **Response to Reviewer N8c9**
> > >
> > > Thank you for your response, which enabled us to address your concerns promptly. We are pleased to offer the following responses to your comments below.
> > >
> > > >**Q1: AAEs with small perturbations**
> > >
> > > **A1:** AAEs do exist in the context of small perturbation-trained SSAT, indicating the presence of non-smooth points within the classifiers. However, in small perturbation-trained SSAT, the outputs' variation between AAE and NAE is tiny, which denotes a small distortion in the classifier and does not appear as CO.
> > >
> > > >**Q2: FGSM loss greater than PGD**
> > >
> > > **A2:** This is indeed an interesting discovery, we observe this phenomenon not only in AAER and SSAT-trained methods but also in multi-step AT-trained methods (e.g., PGD-trained). We further decompose the attack loss into two components: attack success (model defence failure) and attack failure (model defence success) adversarial examples. We found that the mismatch between the average attack loss and the attack success rate is primarily due to the FGSM successful attack adversarial examples which results in significantly higher loss values. Specifically, within the subset of FGSM attack failures  (regardless of whether the PGD attack success), we notice that the attack loss of PGD (loss = 0.5037) is greater than that of FGSM  (loss = 0.4057), indicating that PGD provides a stronger fine-grained attack. However, in the subset of FGSM attacks success, the attack loss of FGSM (loss = 2.1668) is obviously larger than PGD (loss = 1.3832). We speculate that this phenomenon could be attributed to the FGSM's coarse-grained perturbation step, which is particularly effective when dealing with certain straightforward samples.
> > >
> > > If you have any further questions or concerns, please do not hesitate to contact us. We are willing and available to provide any additional information or clarification you may need.

---

> > > > ### Author Response · Authors · 2023-08-21
> > > >
> > > > Thank you once again for your valuable comments, which have helped us improve the clarity and rigour of our manuscript. Please know that we look forward to addressing any further concerns or questions regarding our work.

---

### Official Review · Reviewer_GeQY · 2023-07-07

**Soundness:** 3 good
**Presentation:** 4 excellent
**Contribution:** 3 good
**Rating:** 4
**Confidence:** 3

**Summary:**

This paper introduces the concept of abnormal adversarial examples (AAEs) and shows their strong correlation with the issue of catastrophic overfitting in single-step adversarial training. Based on this observation, the authors propose a new regularization method, AAER, which constrains the generation of abnormal adversarial examples to address this problem. The paper's insights, such as the sudden onset of catastrophic overfitting, are interesting, and the regularization-based approach is flexible enough to be combined with other methods. In addition, the experimental results confirm the effectiveness of the proposed method.

**Strengths:**

1. This paper provides a new perspective to explain the issue of catastrophic overfitting in single-step adversarial training.

2. Some observations in this paper on anomalous adversarial examples are interesting and may provide inspiration for subsequent works.

3. The proposed method is concise and intuitive, and it aligns well with the motivation of the paper.

4. This paper is well writen and easy to follow.

**Weaknesses:**

1. Some claims in this paper are debatable.

2. Insufficient ablation studies.

3. The experimental comparison is insufficient.

More detailed comments please refer to the questions.

**Questions:**

A question of my concern is whether anomalous adversarial examples lead to catastrophic overfitting or in turn, catastrophic overfitting leads to anomalous adversarial samples or; as shown in Figure 2 in this paper, anomalous adversarial examples are also more likely to arise in a model in which the discriminant boundary is highly unsmoothed (i.e., already catastrophically overfitted); if the later, then there is no fundamental difference between this paper and other methods that regularize the smoothness of the discriminant boundary;

The ablation studies in this paper are inadequate, e.g., the baselines of this paper are N-FGSM and RS-FGSM, two advanced single-step adversarial training method, and although the regularization term proposed in this paper improves the robustness of the model even further, it is of interest to know whether the regular approach proposed in this paper is superior to other regular-based methods.

This paper presents some interesting insights in Fig.1 and Fig. 3; however, the comparisons of these experiments were performed in RS-FGSM and RS-AAER, and thus the findings may not reflect the general findings in single-step adversarial training; given that these experimental results are crucial to validate the motivation of this paper, they need to be replicated in vanilla-FGSM and vanilla-AAER for repeated validation.

**Limitations:**

N.A.

---

> ### Author Rebuttal · Authors · 2023-08-10
>
> We sincerely appreciate your time and effort in reviewing our manuscript. Following thorough consideration, please find the responses to your comments below.
>
> >**Q1: Relationship between AAEs and CO**
>
> **A1:** We have observed the presence of AAEs before the occurrence of CO, indicating that CO is not the cause of AAE generation. On the other hand, directly removing AAEs can only delay but not prevent the occurrence of CO, indicating that AAEs are also not the direct cause of CO. The presence of AAEs is correlated with classifier distortion, while CO is a manifestation of a highly distorted classifier. Therefore, hindering the generation of AAEs can be considered as implicitly smoothing the loss landscape, which aligns with the common objective of AT. Notably, from the novel perspective of AAEs, our work provides two valuable findings and proposes an effective method. Firstly, we analyze the CO process and reveal that the classifier already exhibits slight distortions before CO. Secondly, we demonstrate that directly optimizing these AAEs accelerates the distortion process, leading to the rapid manifestation of CO. Finally, a lot of SSAT methods still suffer from CO when faced with strong adversaries (as shown in Tab.1 and A2) or significantly increase the computational overhead (Tab.3). In contrast, our method achieves both efficiency and robustness by explicitly hindering the generation of AAEs.
>
> >**Q2: Compare with other regular-based methods**
>
> **A2:** We compare our method with other regular-based methods, including GAT [1], NuAT [2], PGI-FGSM [3] and Grad Align [4]. The results for GAT, NuAT, and Grad Align are directly taken from [5], while the reported PGI-FGSM results are based on the official code after searching the hyperparameters across different noise magnitudes. We report both natural and **robust** accuracy (divided by a slash) of the final model on the CIFAR10 dataset.
>
> |Method / Noise Magnitude|8/255|12/255|16/255|
> |:-----|:----:|:----:|:----:|
> |GAT [1]|76.75/**50.98**|80.44/**14.93**|82.17/**1.25**|
> |NuAT [2]|73.22/**50.10**|71.15/**17.54**|80.10/**3.29**|
> |PGI-FGSM [3]|77.94/**52.86**|83.82/**5.19**|60.37/**4.16**|
> |Grad Align [4]|81.90/**48.14**|73.29/**34.51**|61.30/**26.64**|
> |RS-AAER|83.83/**46.14**|74.38/**32.17**|64.56/**23.87**|
> |N-AAER|80.56/**48.31**|71.15/**36.52**|61.84/**28.20**|
>
> Based on the above table, we can observe that GAT, NuAT and PGI-FGSM demonstrate superior performance under 8/255 noise magnitude. However, it becomes apparent that these methods still suffer from CO with strong adversaries, resulting in nearly zero robustness under 16/255 noise magnitude, let alone the more challenging 32/255. In contrast, Grad Align and our proposed method consistently exhibit effectiveness across different settings, demonstrating the reliability in preventing CO. Notably, our method performs comparably or even better than Grad Align while requiring only one-third of the training cost (Tab.3). Therefore, compared to other regularization-based methods, AAER achieves both efficiency and robustness in SSAT.
>
> >**Q3: Validate the motivation**
>
> **A3:** In Appendix A, we have demonstrated that by hindering the generation of AAEs, Vanilla-AAER can effectively prevent CO, which indirectly validates our motivation. To further validate our findings, we replicated the experiments on Vanilla-FGSM and Vanilla-AAER, observing consistent phenomena and tendencies as depicted in Fig.1 and Fig.3. We will include these additional experimental results in the revised manuscript.
>
> If you have any further questions or concerns, please do not hesitate to contact us. We are willing and available to provide any additional information or clarification you may need.
>
> [1] Sriramanan et al. "Guided adversarial attack for evaluating and enhancing adversarial defenses." NeuRIPS 2020.
>
> [2] Sriramanan et al. "Towards efficient and effective adversarial training." NeuRIPS 202.
>
> [3] Jia et al. "Prior-guided adversarial initialization for fast adversarial training." ECCV 2022.
>
> [4] Maksym Andriushchenko et al. "Understanding and improving fast adversarial training." NeuRIPS 2020.
>
> [5] de Jorge Aranda et al. "Make some noise: Reliable and efficient single-step adversarial training." NeuRIPS 2022.

---

> ### Author Response · Authors · 2023-08-17
> **Response to Reviewer GeQY**
>
> We hope this message finds you well. We express our sincere appreciation for the time and effort you have dedicated to reviewing our manuscript. As a reminder, we have submitted a rebuttal to address your concerns. In the rebuttal, we explain the relationship between AAE and CO and provide additional experiments to support our method. We understand that the discussion period can be tight, and your prompt response would enable us in addressing any additional concerns. We are looking forward to hearing from you soon.
>
> Best regards,
>
> The Authors

---

### Author Rebuttal · Authors · 2023-08-10

We sincerely appreciate your time and effort in reviewing our manuscript. Following thorough consideration, please find the general responses to your comments below.

>**Q1: More competing baselines**

**A1:** We compare our method with other competing SSAT methods, including GAT [1], NuAT [2], PGI-FGSM [3], SDI-FGSM [4] and Kim [5]. The results for GAT, NuAT and Kim are directly taken from [6], while the reported PGI-FGSM and SDI-FGSM results are based on the official code after searching the hyperparameters across different noise magnitudes. We report both natural and **robust** accuracy (divided by a slash) of the final model on the CIFAR10 dataset.

|Method / Noise Magnitude|8/255|12/255|16/255|
|:-----|:----:|:----:|:----:|
|GAT [1]|76.75/**50.98**|80.44/**14.93**|82.17/**1.25**|
|NuAT [2]|73.22/**50.10**|71.15/**17.54**|80.10/**3.29**|
|PGI-FGSM [3]|77.94/**52.86**|83.82/**5.19**|83.42/**4.16**|
|SDI-FGSM [4]|79.28/**49.26**|70.07/**37.56**|81.09/**0.05**|
|Kim [5]|89.02/**33.01**|88.35/**13.11**|90.45/**1.88**|
|RS-AAER|83.83/**46.14**|74.38/**32.17**|64.56/**23.87**|
|N-AAER|80.56/**48.31**|71.15/**36.52**|61.84/**28.20**|

Based on the above table, we can observe that GAT, NuAT, PGI-FGSM and SDI-FGSM demonstrate superior performance under 8/255 noise magnitude. However, it becomes apparent that these methods still suffer from CO with strong adversaries, resulting in nearly zero robustness under 16/255 noise magnitude, let alone the more challenging 32/255. In contrast, our proposed method exhibits consistent effects across different settings with negligible computational overhead, demonstrating its trustworthy effectiveness in preventing CO. We would like to emphasize that while achieving excellent performance under 8/255 noise magnitude is certainly gratifying, but can reliable defence against CO is more critical to a successful SSAT method.

>**Q2: Computational overhead**

**A2:** The AAER method does not involve any extra generation, forward, or backward propagation processes, all additional processing is solely based on the original FGSM intermediate variables, which makes it highly efficient. The training times for N-AAER and RS-AAER are nearly identical, and for clarity, we consolidate them into AAER (30.5S), as depicted in Section 4.4 "Time Complexity Study". In terms of computational overhead, our method demonstrates a comparable cost, with only a 16% increase compared to the RS-FGSM (26.1S). In contrast, Grad Align (69.4S) and PGD-10 (140.7S) exhibit significantly higher training times, being 2.3 and 4.6 times slower than our proposed method, respectively. **Additionally, we are delighted to report that we have observed a minimal 1.8% increase in RS/N-AAER time overhead compared to RS/N-FGSM. This observation was made using a single NVIDIA RTX 4090 GPU with PyTorch 2.0, and all experiments without half-precision float.**

|Method|FreeAT|ZeroGrad|MultiGrad|Grad Align|RS\N-FGSM|RS\N-AAER|PGD-2|PGD-10|
|:-----|:----:|:----:|:----:|:----:|:----:|:----:|:----:|:----:|
|Training Time (S)|43.8|11.0|21.7|36.1|11.0|11.2|16.4|59.1|
|Ratio (%)|391.07|98.21|193.75|322.32|98.21|100|146.42|527.67|

**NOTE: AAER as a baseline for training time ratio.**

If you have any further questions or concerns, please do not hesitate to contact us. We are willing and available to provide any additional information or clarification you may need.

[1] Sriramanan et al. "Guided adversarial attack for evaluating and enhancing adversarial defenses." NeuRIPS 2020.

[2] Sriramanan et al. "Towards efficient and effective adversarial training." NeuRIPS 2021.

[3] Jia, X et al. "Prior-guided adversarial initialization for fast adversarial training." ECCV 2022.

[4] Jia et al. "Boosting fast adversarial training with learnable adversarial initialization." IEEE Transactions on Image Processing, 2022.

[5] Kim et al. "Understanding catastrophic overfitting in single-step adversarial training." AAAI 2021.

[6] de Jorge Aranda et al. "Make some noise: Reliable and efficient single-step adversarial training." NeuRIPS 2022.

---

### Author Response · Authors · 2023-08-15
**Rolling Discussion Invitation**

Dear reviewers,

We have diligently addressed most of the concerns during the rebuttal period. We would like to invite reviewers that still have concerns about our manuscript to join the rolling discussion, where we are looking forward to receiving your valuable insights and comments.

Best regards,

The Authors

---

### Comment · Area_Chair_KSvq · 2023-08-17
**Waiting for Reviewers' Feedback**

Dear Reviewer GeQY, N8c9, EF7n, and 6vch,

The authors and I are eager to know whether the author responses successfully address your concern. The Author-Reviewer discussion ends on **August 21**. You are strongly encouraged to directly reply to the authors.

Thank you for your hard work.

PS: (1) This is a public thread. (2) I am expecting to know your thoughts as well. If you want to individually reply to me, please use the thread of internal discussion.

Kind Regards,

AC

---

### Decision · Program_Chairs · 2023-09-21

**Decision:**

Accept (poster)

**Comment:**

This is the second time to review this paper as an AC. The current version has significant improvements and addressed most of my concerns in the last round. I have reviewed this paper again by myself and my initial evaluation on this paper is "accept" in terms of improved technique and personal emotion.

**Research Problem**

The authors study the catastrophic overfitting problem, that classifier decision boundaries are highly distorted, and robust accuracy against iterative-step adversarial attacks suddenly drops from peak to nearly 0% in a few epochs. Specifically, the authors noticed some adversarial examples generated on the network with distorted decision boundaries exhibit anomalous behavior, which co-occurs with catastrophic overfitting. In light of this, the authors aim to prevent catastrophic overfitting by suppressing generated abnormal adversarial examples and designing a regularizer.

**Motivation**

Personally, I like that the motivation is derived from some empirical findings.

**Philosophy**

This paper starts from the conjecture that catastrophic overfitting might result from abnormal adversarial examples. Based on that, the authors design a regularizer to suppress the generation of abnormal adversarial examples.

**Presentation**

This is the second time to review this paper. The current version has significant improvement on the presentation.

**Technique**

The novelty of this paper is the proposed Abnormal Adversarial Examples Regularization. The technical roadmap that combines the definition of NAE and AAE, empirical validation and analyses, makes sense to me.

**Experiments**

1. The experimental results are extensive.

2. To increase the readability, “S” can be removed from the table, but it can be mentioned in the table caption, such as “training time by second.”

3. It is nice to see that the authors involve more other regularizers that tackle catastrophic overfitting, according to the comments from the previous submission.

4. Figure 5 (a) is the new content that demonstrates the effectiveness of reducing the number of abnormal adversarial examples, which is good to address the comments from the previous submission.

**Clarification on My Comments**

1. The scores from reviewers are the initial ones before the author response. Due to the no participation of reviewers during the internal discussion, these scores were unchanged.

2. I have checked the comments of reviewers. Some of them are new and good compared to the last review I had. I also checked the author response, which was helpful to address these concerns. I **strongly** suggest the authors to modify their manuscript in the camera-ready according to the reviewers' comments, if accepted. For example, whether anomalous adversarial examples lead to catastrophic overfitting or in turn, catastrophic overfitting leads to anomalous adversarial samples.

3. I have discussed with SAC. SAC supported my recommendation. I was very thankful for the support from SAC.